# Ataxin-3 consolidates the MDC1-dependent DNA double-strand break response by counteracting the SUMO-targeted ubiquitin ligase RNF4

Annika Pfeiffer[1,†], Martijn S Luijsterburg[2,†], Klara Acs[1], Wouter W Wiegant[2], Angela Helfricht[2], Laura K Herzog[1], Melania Minoia[1], Claudia Böttcher[1], Florian A Salomons[1], Haico van Attikum[2,*] & Nico P Dantuma[1,**]

## Abstract

The SUMO-targeted ubiquitin ligase RNF4 functions at the crossroads of the SUMO and ubiquitin systems. Here, we report that the deubiquitylation enzyme (DUB) ataxin-3 counteracts RNF4 activity during the DNA double-strand break (DSB) response. We find that ataxin-3 negatively regulates ubiquitylation of the checkpoint mediator MDC1, a known RNF4 substrate. Loss of ataxin-3 markedly decreases the chromatin dwell time of MDC1 at DSBs, which can be fully reversed by co-depletion of RNF4. Ataxin-3 is recruited to DSBs in a SUMOylation-dependent fashion, and *in vitro* it directly interacts with and is stimulated by recombinant SUMO, defining a SUMO-dependent mechanism for DUB activity toward MDC1. Loss of ataxin-3 results in reduced DNA damage-induced ubiquitylation due to impaired MDC1-dependent recruitment of the ubiquitin ligases RNF8 and RNF168, and reduced recruitment of 53BP1 and BRCA1. Finally, ataxin-3 is required for efficient MDC1-dependent DSB repair by non-homologous end-joining and homologous recombination. Consequently, loss of ataxin-3 sensitizes cells to ionizing radiation and poly(ADP-ribose) polymerase inhibitor. We propose that the opposing activities of RNF4 and ataxin-3 consolidate robust MDC1-dependent signaling and repair of DSBs.

**Keywords** deubiquitylation enzyme; DNA damage response; DNA repair; SUMO; ubiquitin
**Subject Categories** DNA Replication, Repair & Recombination; Post-translational Modifications, Proteolysis & Proteomics
**The EMBO Journal (2017) 36: 1066–1083**

## Introduction

Ubiquitylation of histones and chromatin-associated proteins plays a pivotal role in coordinating the cellular response to DNA double-strand breaks (DSBs; Dantuma & van Attikum, 2016). A large number of ubiquitin ligases mediates the ubiquitin-dependent recruitment of DNA damage response proteins to DSBs. The ubiquitin ligase RNF8, which is recruited to DSBs by interacting with ATM-phosphorylated MDC1, marks chromatin-associated proteins in the vicinity of sites of DNA damage with ubiquitin (Huen *et al*, 2007; Kolas *et al*, 2007; Mailand *et al*, 2007). In particular, ubiquitylation of linker histone H1 by RNF8 facilitates the sequestration of the ubiquitin ligase RNF168 (Thorslund *et al*, 2015), which initiates robust DNA damage-induced ubiquitylation of chromatin surrounding DSBs, thereby recruiting DNA damage response proteins such as 53BP1 and BRCA1 (Mattiroli *et al*, 2012).

The ubiquitin-like modifier SUMO is also critically engaged in the regulation of the DSB response (Jackson & Durocher, 2013). Although a variety of chromatin-associated proteins involved in this process are modified by the SUMO ligases PIAS1 and PIAS4 (Galanty *et al*, 2009; Morris *et al*, 2009), the molecular significance of SUMOylation in response to DNA damage is not fully understood. It has been proposed that protein-group SUMOylation functions in yeast as a protein glue that stabilizes protein interactions on chromatin (Psakhye & Jentsch, 2012). At the same time, more specific effects have been reported for BRCA1 for which SUMOylation has been shown to enhance the ubiquitin ligase activity of the BRCA1/BARD1 complex (Morris *et al*, 2009). Recent studies demonstrated that DNA damage-induced SUMOylation also results in the recruitment of the SUMO-targeted ubiquitin ligase (STUbL) RNF4 to DNA lesions, where it facilitates the ubiquitin-dependent removal of chromatin-associated MDC1 and RPA, which was suggested to promote DSB repair by non-homologous end-joining (NHEJ) and homologous recombination (HR), respectively (Galanty *et al*,

1 Department of Cell and Molecular Biology, Karolinska Institutet, Stockholm, Sweden
2 Department of Human Genetics, Leiden University Medical Center, Leiden, The Netherlands
  *Corresponding author. Tel: +31 71 5269624; E-mail: h.van.attikum@lumc.nl
  **Corresponding author. Tel: +46 8 52487384; E-mail: nico.dantuma@ki.se
  † These authors contributed equally to this work

2012; Luo *et al*, 2012; Yin *et al*, 2012). Whether activities exist that regulate the actions of RNF4 at chromatin is currently unclear.

The understanding that deubiquitylation enzymes (DUBs) play an important role in regulating ubiquitin-dependent processes by virtue of their ability to selectively disassemble ubiquitin chains has gained a lot of interest during recent years (Komander *et al*, 2009). The DUB ataxin-3, which is the founding member of the family of Josephin proteases (Burnett *et al*, 2003), disassembles canonical K48-linked chains, as well as non-proteolytic K63-linked ubiquitin chains, suggesting diverse roles of this DUB in ubiquitin signaling events (Todi *et al*, 2009, 2010; Winborn *et al*, 2008). Although its biochemical properties are relatively well understood, studies on the biological functions of ataxin-3 give a more diffuse picture, suggesting regulatory roles of cytosolic ataxin-3 in protein quality control (Burnett & Pittman, 2005; Wang *et al*, 2006; Zhong & Pittman, 2006). Nuclear ataxin-3, on the other hand, has been shown to interact with chromatin and is involved in repressing transcription by interacting with histone deacetylases (Evert *et al*, 2006). In addition, it regulates the activity of polynucleotide kinase 3′-phosphatase, which is a DNA-processing enzyme that is involved in the repair of DNA single-strand breaks (Chatterjee *et al*, 2015). Together, these studies suggest that ataxin-3 has pleiotropic effects on cellular physiology by regulating both cytosolic and nuclear processes.

In the present study, we show that ataxin-3 has an important regulatory function in the activation of the DSB response. Ataxin-3 is recruited to DSBs independently of its ubiquitin-interacting motifs (UIMs), but instead relies on DNA damage-induced SUMOylation. Our data suggest that ataxin-3 behaves as a DUB that counteracts the STUbL RNF4 at DSBs, thereby preventing premature removal of MDC1, which, in turn, ensures efficient recruitment of DSB response factors, such as RNF8, RNF168, BRCA1, and 53BP1. We also show that ataxin-3 promotes efficient MDC1-dependent DSB repair by NHEJ and HR. Our findings suggest that the opposing activities of RNF4 and ataxin-3 consolidate robust MDC1-dependent signaling and repair of DSBs.

# Results

### Ataxin-3 is recruited to DSBs

It has been previously reported that extraction of chromatin-associated proteins by the ubiquitin-selective segregase valosin-containing protein VCP/p97 plays an important role in the DSB response (Acs *et al*, 2011; Meerang *et al*, 2011). Ubiquitin-dependent extraction from chromatin (Dantuma & Hoppe, 2012) resembles the ubiquitin-dependent extraction of aberrant proteins from the endoplasmic reticulum (ER) membrane prior to protea-somal degradation (Meyer *et al*, 2012), a process in which the DUB ataxin-3 has been shown to assist VCP/p97 (Wang *et al*, 2006). Inspired by the link between VCP/p97 and ataxin-3 in ER-associated degradation, we decided to explore a possible role of ataxin-3 in the DSB response. Given that VCP/p97 is recruited to sites of DNA damage (Acs *et al*, 2011; Meerang *et al*, 2011), we asked whether ataxin-3 also accumulates at these sites. To this end, we inflicted DSBs with a pulsed UV-A (365 nm) laser in

human cells that had been sensitized with 5′-bromo-2-deoxyuri-dine (BrdU). Microscopic analysis revealed that endogenous ataxin-3 (Fig 1A) and ectopically expressed GFP-ataxin-3 (Fig 1B) are recruited to DNA damage. In a second approach, we used U2OS-DSB reporter cells in which DSBs can be introduced by inducible tethering of an mCherry-tagged fusion of the bacterial LacR protein and the nuclease FokI (mCherry-LacR-FokI) to a stably integrated LacO array (Tang *et al*, 2013). Chromatin-tethered mCherry-LacR-FokI resulted in the generation of DSBs marked by phosphorylated H2AX (γH2AX) to which GFP-ataxin-3 was recruited, confirming the DNA damage-induced translocation of ataxin-3 to *bona fide* DSBs (Fig 1C). The recruitment of ataxin-3 to DSBs is consistent with a recently reported systematic charac-terization of DUBs, which showed that ataxin-3 accumulates at laser-inflicted DNA damage (Nishi *et al*, 2014).

Given that ataxin-3 and VCP/p97 are known to act cooperatively in ER-associated degradation (Meyer *et al*, 2012), we investigated whether their recruitment to DSBs is interdependent. The interaction between ataxin-3 and VCP/p97 has been well characterized and depends on an intrinsic VCP/p97-binding motif (VBM) in ataxin-3 (Boeddrich *et al*, 2006). While mutating the VBM in ataxin-3 indeed prevented binding of ataxin-3 to VCP/p97 (Appendix Fig S1A), it did not abrogate recruitment of ataxin-3 to sites of DNA damage (Fig 1D and E). The recruitment of VCP/p97 itself was also indepen-dent of ataxin-3, suggesting that the majority of VCP/p97 and ataxin-3 are recruited independently to sites of DNA damage (Appendix Fig S1B).

An alternative explanation for the recruitment of ataxin-3 to DSBs is the presence of ubiquitylated chromatin, which could be recognized by the three UIMs present in ataxin-3. However, simultaneous introduction of mutations in the three UIMs of ataxin-3, which abrogate their ubiquitin-binding properties (Berke *et al*, 2005), did not prevent accumulation of ataxin-3 at laser-inflicted (Fig 1D) or nuclease-induced DSBs (Fig 1E). Moreover, we found that replacing endogenous RNF8 with a catalytically inactive mutant (C403S; denoted as *RING), which impairs RNF8-dependent ubiquitylation and 53BP1 accumulation at DSBs (Mailand *et al*, 2007), did not impair accrual of ataxin-3 (Fig 1F and G). Finally, also the catalytic activity of ataxin-3 was dispensable for its recruitment to DSBs since ataxin-3[C14A], in which the catalytic cysteine residue has been substituted with alanine, also readily localized to sites of DNA damage (Fig 1D and E). Together our data show that ataxin-3 is recruited to DSBs independently of its ubiquitin- and VCP/p97-binding domains or its catalytic activity.

### Ataxin-3 is recruited to DSBs by DNA damage-induced SUMOylation

We noticed that DNA damage response proteins accumulate in distinct spatial patterns at FokI nuclease-induced DSBs. Notably, ubiquitin conjugates (Fig 2A), VCP/p97 (Fig 2B), and 53BP1 (Fig 2C) were enriched over a region that surpasses the actual sites of DSBs marked by the mCherry-LacR-FokI fusion where they co-localized with γH2AX or MDC1, consistent with the notion that histone phosphorylation and ubiquitylation are amplified and spread from the actual DSB sites to the surrounding chromatin (Lukas *et al*, 2011). In contrast, GFP-ataxin-3 did not spread to the

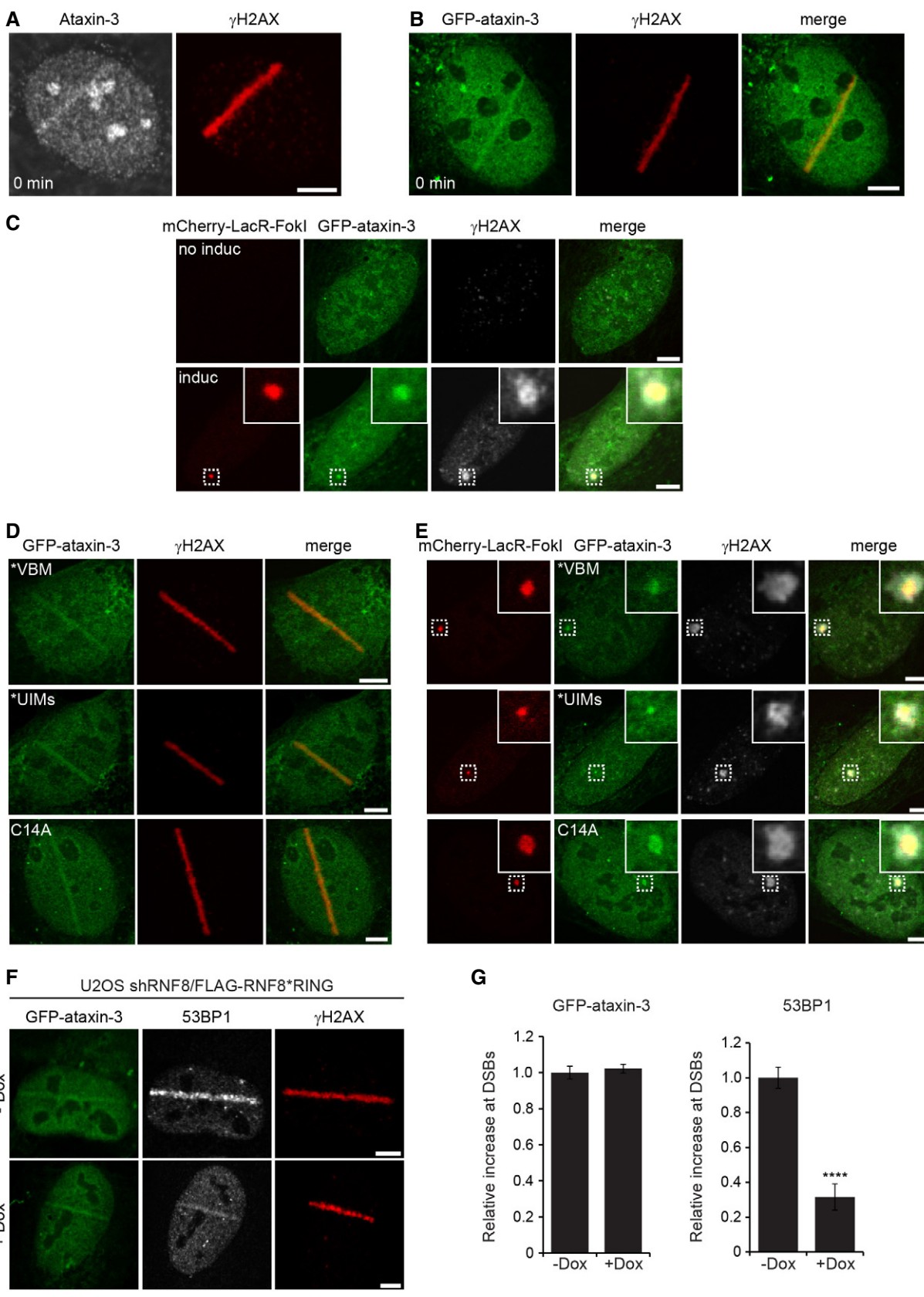

**Figure 1.**

◄  **Figure 1.   Ataxin-3 is recruited to DSBs independently of its ubiquitin- or VCP-binding motifs.**

A, B  U2OS cells (A) or U2OS cells transiently transfected with GFP-ataxin-3 (B) were laser micro-irradiated, directly fixed, and analyzed by immunofluorescence with ataxin-3 and γH2AX antibodies.

C  U2OS-DSB reporter cells were transfected with GFP-ataxin-3 and analyzed by immunofluorescence 5 h after induction of DSBs by mCherry-LacR-FokI. Immunostaining of γH2AX was performed to confirm the induction of DSBs. The magnified areas are indicated.

D  U2OS cells were transiently transfected with GFP-ataxin-3*VBM, GFP-ataxin-3*UIMs, or catalytically inactive GFP-ataxin-3$^{C14A}$. Cells were subjected to laser micro-irradiation, directly fixed, and analyzed by immunofluorescence.

E  U2OS-DSB reporter cells were transfected with GFP-ataxin-3*VBM, GFP-ataxin-3*UIMs, or GFP-ataxin-3$^{C14A}$ and analyzed by immunofluorescence 5 h after induction of DSBs by mCherry-LacR-FokI. Immunostaining of γH2AX was performed to confirm the induction of DSBs. The magnified areas are indicated.

F  U2OS cells stably expressing doxycycline-inducible shRNF8/FLAG-RNF8$^{C403S}$ were induced (1 μg/ml doxycycline; Dox) or not and transfected with GFP-ataxin-3. Laser micro-irradiation was conducted, and cells were analyzed by immunofluorescence.

G  Quantification of the relative increase in 53BP1 (positive control) and GFP-ataxin-3 in U2OS cells stably expressing Dox-inducible shRNF8/FLAG-RNF8$^{C403S}$ or not. Data are presented as mean ± SEM from three independent experiments. ****$P \leq 0.0001$ (Mann–Whitney test).

Data information: Scale bars, 5 μm.

perimeter of the γH2AX foci, but displayed a distribution that was confined to the inner region marked by mCherry-LacR-FokI (Fig 2D). Interestingly, we observed that the confined localization of ataxin-3 corresponded with the distribution of SUMO-1 (Fig 2E) and Ubc9 (Fig 2F), which is the E2 enzyme responsible for SUMOylation, suggesting a possible role for this modification in ataxin-3 recruitment. In addition, we found a similar localization for the SUMO-binding protein RNF4 (Fig 2G). A role for SUMO in ataxin-3 recruitment was supported by the observation that depletion of the SUMO conjugase Ubc9 or the SUMO ligase PIAS4, which are both required for DNA damage-induced SUMOylation (Galanty *et al*, 2009), prevented the recruitment of ataxin-3 to laser-inflicted DNA lesions (Fig 2H and Appendix Fig S2A). This demonstrates that SUMOylation is required for sequestration of ataxin-3 at sites of DNA damage.

To address whether ataxin-3 can associate with SUMO, we used recombinant SUMO1 or SUMO2 conjugated to beads to perform pull-down assays from cell lysates. We observed that endogenous ataxin-3 interacted with SUMO1, whereas hardly any interaction with SUMO2 could be detected under our assay conditions (Fig 3A). Interestingly, the interaction with SUMO1 was also enhanced upon exposure of cells to the DSB-inducing agent bleomycin, indicating that the ability of ataxin-3 to interact with SUMO-1 is stimulated by DNA damage. SUMO1 beads also pulled down recombinant ataxin-3 showing that they can directly interact (Appendix Fig S2B). To identify the domain responsible for the interaction with SUMO1, we used lysates of cells expressing GFP-tagged full-length ataxin-3 or an N- or C-terminal ataxin-3 fragment (Fig 3B). This revealed that the N-terminal Josephin domain, but not the C-terminal domain of ataxin-3, associated with SUMO1 (Fig 3C). It has been reported previously that ataxin-3 contains a putative SUMO-interacting motif (SIM) in its Josephin domain although its functionality as a SUMO-binding module has not been empirically tested (Guzzo & Matunis, 2013). To test the role of the putative SIM, we generated full-length ataxin-3 in which the motif had been mutated ($^{162}$IFVV to $^{162}$AFAA) by three amino acid substitutions. Interestingly, we observed that ataxin-3 carrying a mutated SIM was neither able to interact with SUMO1 (Fig 3D), nor able to accumulate at laser- and nuclease-inflicted DNA damage (Fig 3E), consistent with the idea that the putative SIM interacts with DNA damage-induced SUMO conjugates. In summary, our experiments show that ataxin-3 interacts with SUMO1 and that SUMOylation is required for the recruitment of ataxin-3 to DSBs.

## Ataxin-3 and RNF4 have opposing effects on the chromatin dwell time of MDC1 at DNA damage

We next sought to link the SUMO-dependent recruitment of ataxin-3 to its role as a DUB. To this end, we considered the possibility that ataxin-3 may act upon a SUMO-dependent ubiquitylation event during the DSB response. We focused on RNF4 since it is known that this STUbL acts at the crossroads of SUMO and ubiquitin conjugation systems at DSBs (Galanty *et al*, 2012; Luo *et al*, 2012; Yin *et al*, 2012). We first analyzed whether accumulation of ataxin-3 and RNF4 coincided at DSBs and observed that RNF4, like ataxin-3, showed rapid accumulation at sites of DNA damage (Fig 4A), as has been shown previously (Vyas *et al*, 2013). Thus, the localization of ataxin-3 and RNF4 at DSBs is temporally and spatially similar (see Fig 2G), raising the possibility that they share SUMOylated substrates.

The checkpoint mediator MDC1 is a likely candidate to be such a shared SUMOylated substrate (Galanty *et al*, 2012; Luo *et al*, 2012; Yin *et al*, 2012). Since SUMO-targeted ubiquitylation of MDC1 by RNF4 results in a reduced chromatin retention time of MDC1 at DSBs (Galanty *et al*, 2012), we examined the effect of ataxin-3 depletion (Appendix Fig S3A) on the exchange kinetics of MDC1 from DSB-containing chromatin in living cells. To this end, GFP-MDC1 that had accumulated at laser-inflicted DNA damage was photobleached in roughly half of the damaged area after which both the fluorescence recovery in the bleached area and the fluorescence loss in the non-bleached area were monitored as a readout of the exchange kinetics of MDC1 at damaged sites (Fig 4B). Remarkably, knock-down of ataxin-3 resulted in an almost threefold faster exchange of MDC1, suggesting that ataxin-3 contributes to a more stable retention of MDC1 on DSB-containing chromatin (Fig 4C and D). Real-time imaging of GFP-tagged MDC1 to laser damage confirmed that the enhanced exchange was accompanied by a mild delay in MDC1 recruitment in ataxin-3-depleted cells, which was most striking early after DNA damage induction (Fig 4E). Our findings suggest that MDC1 molecules are not efficiently retained on damaged chromatin in ataxin-3-depleted cells, but rapidly replaced by new MDC1 molecules, explaining the strong impact on the exchange rate and mild effect on the steady-state bound levels of MDC1 at DSBs. Thus, while RNF4 reduces the chromatin retention of MDC1 in wild-type cells (Galanty *et al*, 2012), ataxin-3 has the opposite effect and increases the retention time of MDC1 on chromatin. Notably, depletion of ataxin-3 did not affect the steady-state

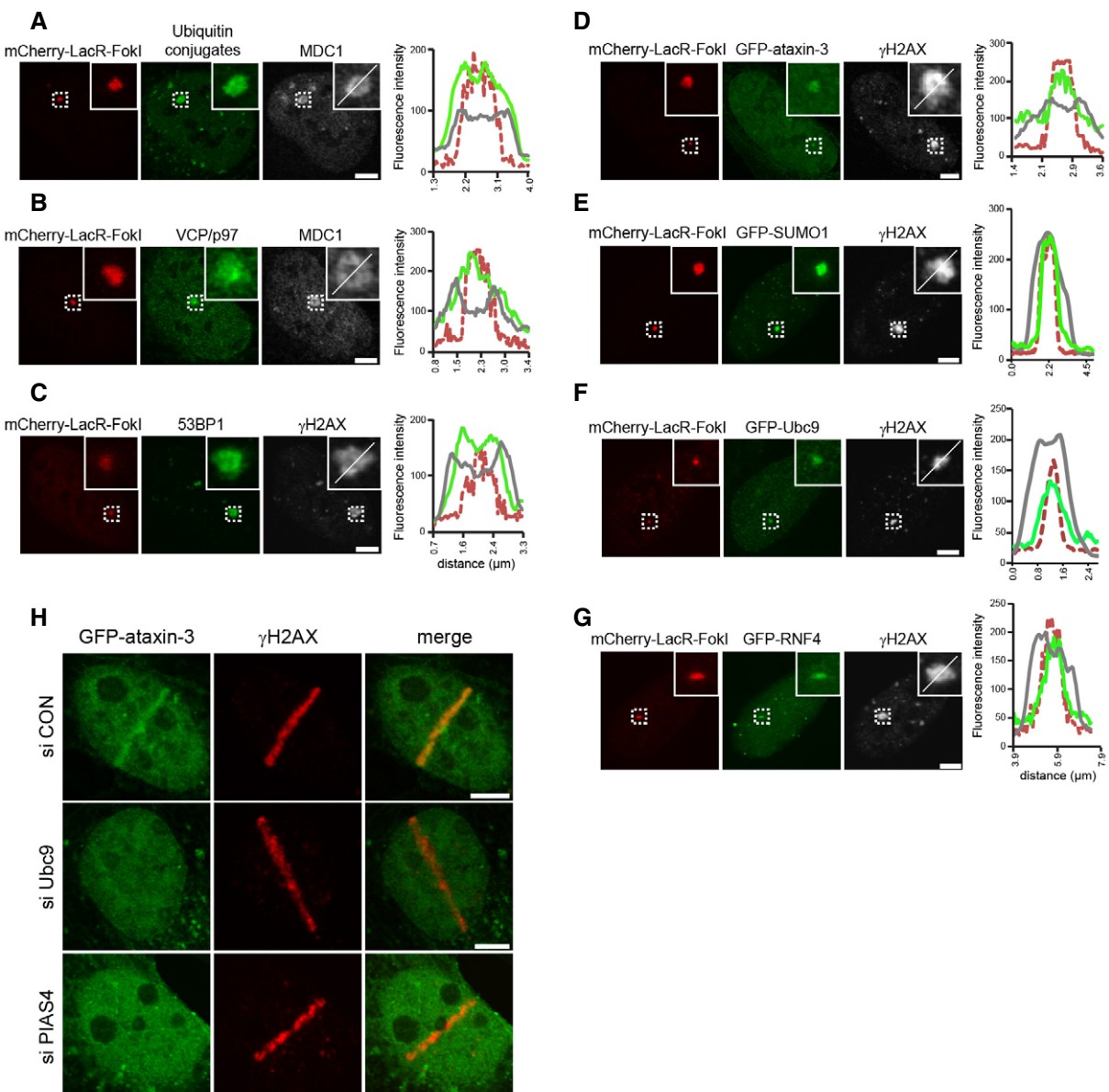

**Figure 2. DNA damage-induced SUMOylation promotes the recruitment of ataxin-3 to DSBs.**

A–G  U2OS-DSB reporter cells were induced by 1 μM 4-OHT and 1 μM Shield-1 for tethering of mCherry-LacR-FokI to the integrated LacO. The localization of (A) ubiquitin conjugates (FK2 antibody) and MDC1, (B) VCP/p97 and MDC1, (C) 53BP1 and γH2AX, (D) GFP-ataxin-3 and γH2AX, (E) GFP-SUMO1 and γH2AX, (F) GFP-Ubc9 and γH2AX, and (G) GFP-RNF4 and γH2AX was determined by native fluorescence (GFP) or immunostaining. Line scans at sites of DSBs were performed to visualize the spatial distribution of indicated proteins. The color coding of the line scans equals the micrographs.

H  GFP-ataxin-3 transfected U2OS cells were treated with control, Ubc9, or PIAS4 siRNAs. Cells were laser micro-irradiated, directly fixed, and analyzed by immunofluorescence.

Data information: Scale bars, 5 μm.

levels of RNF4, suggesting that the reduced MDC1 retention in the absence of ataxin-3 is not an indirect effect due to altered levels of RNF4 (Appendix Fig S3B). To test whether RNF4 and ataxin-3 act epistatically on the MDC1 exchange rate, we performed double knock-down experiments. Indeed, co-depletion of RNF4 restored the exchange rate of MDC1 in ataxin-3-depleted cells to that observed in control cells, suggesting that ataxin-3 affects MDC1 exchange by acting on RNF4-ubiquitylated MDC1 (Fig 4C and D).

**Ataxin-3 and RNF4 regulate ubiquitylation of MDC1**

To further explore the link between ataxin-3 and MDC1, we performed co-immunoprecipitation experiments. These revealed an interaction between endogenous MDC1 and endogenous ataxin-3, which was not enhanced by DNA damage (Fig 5A). Moreover, we could still detect this interaction in cells depleted of Ubc9 or PIAS4 (Appendix Fig S4A), or when using an MDC1 mutant that lacks the

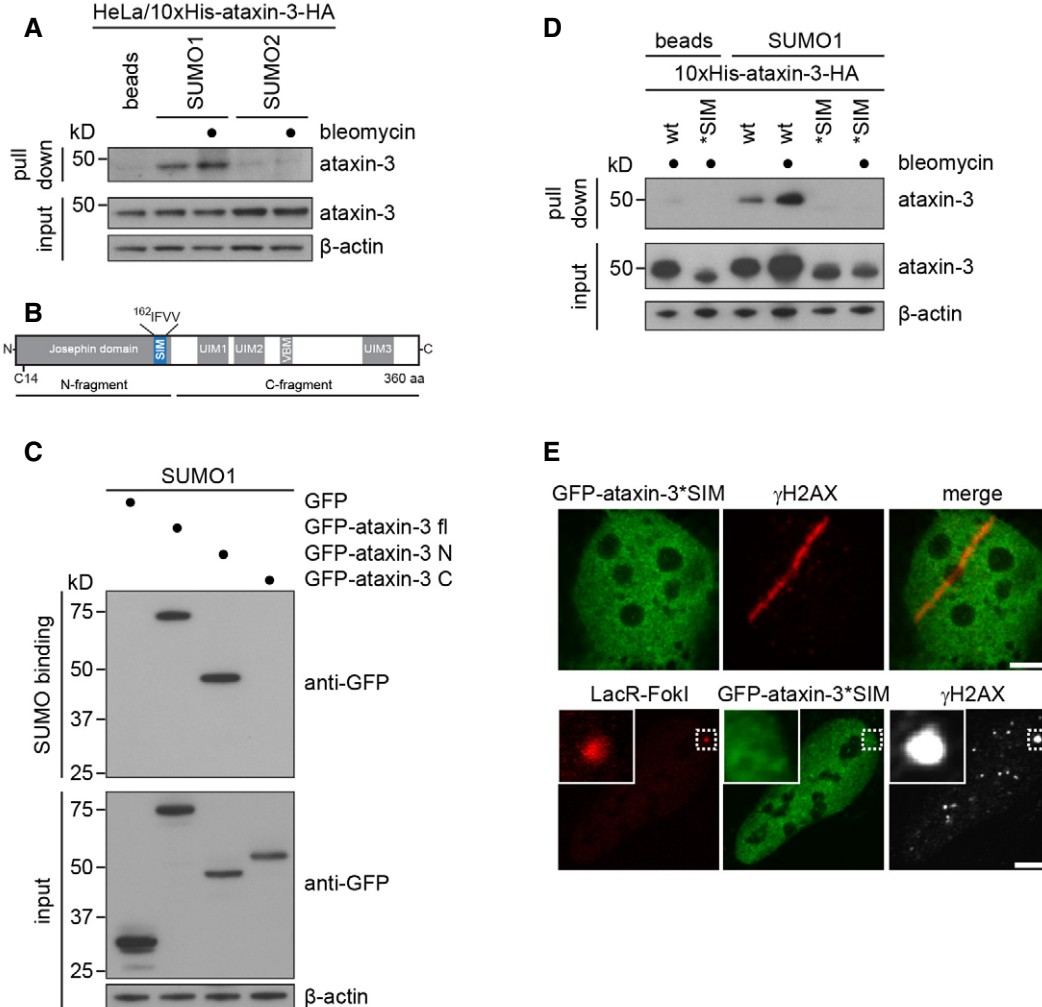

**Figure 3. Ataxin-3 interacts with SUMO1.**

A   Lysates from HeLa cells expressing Dox-inducible 10xHis-ataxin-3-HA that had been treated with 10 μg/ml bleomycin or left untreated were incubated with unconjugated agarose beads or with recombinant SUMO1 or SUMO2 immobilized on agarose beads. Binding of ataxin-3 to recombinant SUMO was analyzed by immunoblotting.

B   Schematic drawing of ataxin-3, which contains the catalytic Josephin domain (N-terminal fragment) and ubiquitin-interacting and VCP/p97-binding motifs (UIMs and VBM, respectively) in its C-terminal fragment.

C   SUMO1 binding assay from U2OS cell lysates transfected with GFP, full-length GFP-ataxin-3, or the GFP-tagged N- or C-terminal fragments of ataxin-3.

D   SUMO1 binding was performed comparing wild-type or putative SIM mutated 10xHis-ataxin-3-HA.

E   Ataxin-3 harbors a consensus SUMO-interacting motif (SIM) in its Josephin domain ($^{162}$IFVV), see (B). The recruitment of GFP-ataxin-3*SIM to DSBs in U2OS cells was analyzed by laser micro-irradiation or in U2OS-DSB reporter cells expressing mCherry-LacR-FokI.

Data information: Scale bars, 5 μm.

major SUMOylation site (K1840) (Appendix Fig S4B). Finally, depletion of MDC1 did not substantially reduce recruitment of ataxin-3 to DSBs, suggesting that the SUMO-dependent recruitment of ataxin-3 is not solely dependent on MDC1 (Appendix Fig S4C), similar to what has been observed for RNF4 (Galanty *et al*, 2012). These findings suggest that the interaction between ataxin-3 and MDC1 is constitutive and does not require the DNA damage-induced SUMOylation of MDC1.

We next sought to address the relevance of the MDC1-ataxin-3 interaction by monitoring the ubiquitylation status of MDC1 in ataxin-3 knock-down cells. To this end, we transfected cells with HA-tagged ubiquitin and treated them with the DNA-damaging agent camptothecin to induce ubiquitylation of MDC1. Strikingly, we found that depletion of ataxin-3 strongly increased the levels of polyubiquitylated MDC1 (Fig 5B). Corroborating these findings, we also detected a substantial increase in the levels of polyubiquitylated MDC1 with ubiquitin-specific antibodies in ataxin-3-depleted cells after treatment with the DNA-damaging agent bleomycin. Importantly, this phenotype could be rescued by ectopic expression of wild-type ataxin-3, but not catalytically inactive ataxin-3$^{C14A}$ (Fig 5C). These findings suggest that MDC1 is a substrate of the deubiquitylating activity of ataxin-3. To examine

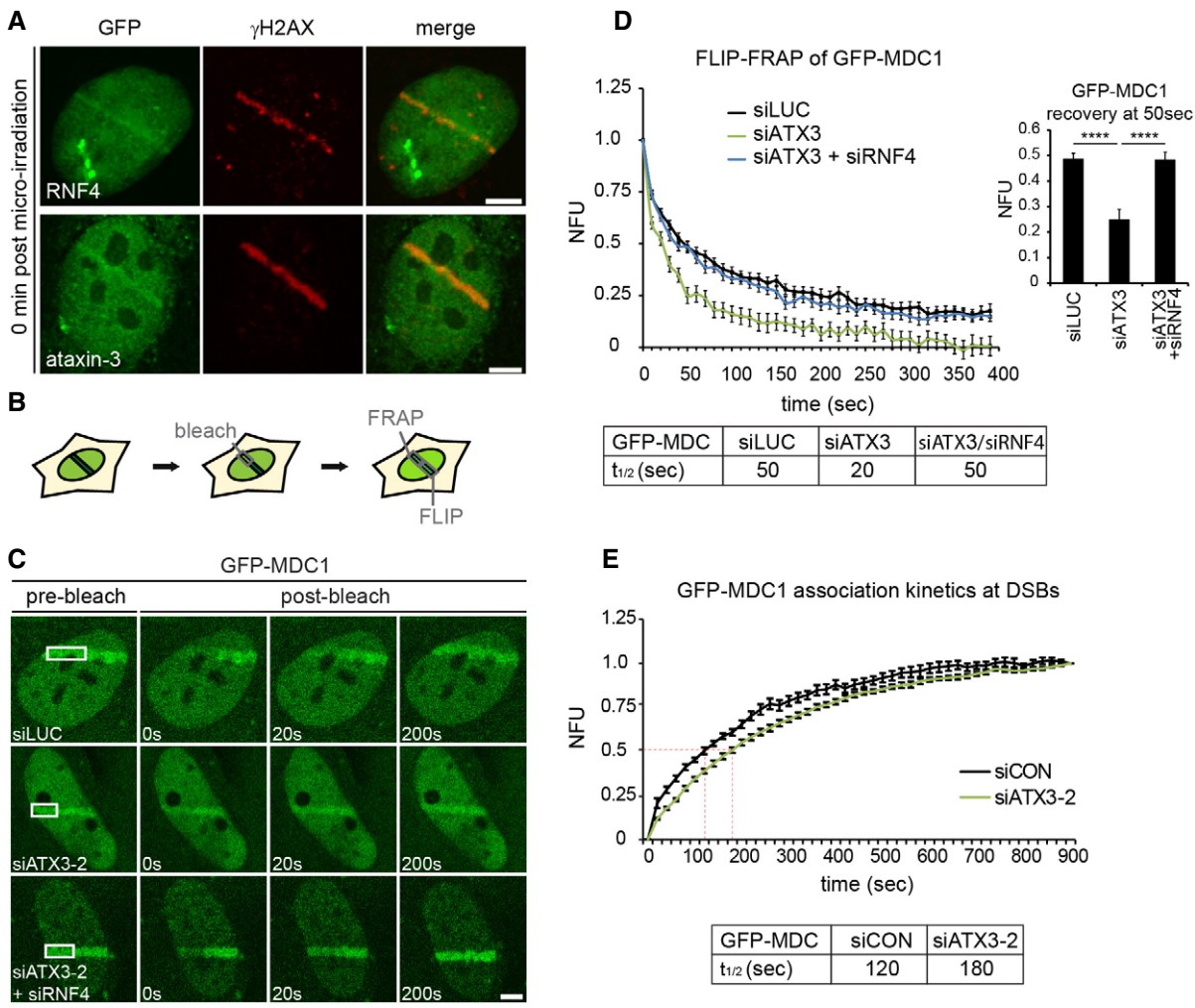

**Figure 4. Ataxin-3 and RNF4 have opposing effects on the chromatin exchange of MDC1 at DNA damage.**

A  U2OS cells transfected with GFP-RNF4 or GFP-ataxin-3 were subjected to laser micro-irradiation, fixed immediately, and analyzed by immunolabeling.

B  U2OS cells stably expressing GFP-MDC1 were laser micro-irradiated, and GFP-MDC1 was allowed to accumulate at laser-induced DNA damage for 3 min. One half of the laser tracks was photobleached. The loss of fluorescence (FLIP) in the non-bleached half and the recovery of fluorescence (FRAP) in the bleached half were monitored with low laser power.

C  Micrographs of GFP-MDC1 at sites of DNA damage in micro-irradiated living cells. The pre-bleach and indicated post-bleach time points are shown.

D  Quantification of the exchange rate of GFP-MDC1 is represented as FLIP-FRAP normalized to 1. The recovery time of GFP-MDC1 in bleached areas at 50 s in control, ataxin-3-depleted, or ataxin-3- and RNF4-depleted cells is depicted. Data are presented as mean ± SEM from two independent experiments. ****$P \leq 0.0001$ (Kruskal–Wallis test).

E  Quantification of time lapse imaging of GFP-MDC1 in living cells. Stably expressing GFP-MDC1 U2OS cells were transfected with indicated siRNAs, subjected to laser micro-irradiation, and monitored for 15 min under live cell imaging conditions, taking an image every 20 s. Accumulated levels of GFP-MDC1 on laser lines were determined, and $t_{1/2}$ was calculated. Data are presented as mean ± SEM of two independent experiments.

Data information: Scale bars, 5 μm.

whether RNF4 and ataxin-3 have opposed activities in regulating MDC1 ubiquitylation, we performed single and double knock-down experiments. While knock-down of ataxin-3 resulted in increased ubiquitylated MDC1, knock-down of RNF4 reduced the levels of ubiquitylated MDC1 in control and ataxin-3-depleted cells (Fig 5D). This suggests that ataxin-3 acts upon ubiquitin chains on MDC1 catalyzed by RNF4. Together these findings indicate that ataxin-3 negatively regulates the RNF4-dependent ubiquitylation of MDC1, providing an explanation for how ataxin-3 increases the residence time of MDC1 at DSBs.

Even though the MDC1-ataxin-3 interaction was not enhanced by DNA damage, we did find that SUMO1 interacts with the catalytic domain of ataxin-3 (see Fig 3C). Considering that MDC1 is specifically SUMOylated after DNA damage and is a substrate of ataxin-3, we asked whether SUMO1 affects the deubiquitylating activity of ataxin-3. Interestingly, we observed that pre-incubating recombinant ataxin-3 with SUMO1 significantly increased its activity toward ubiquitin chains *in vitro* (Fig 5E). These findings suggest ataxin-3 may display a stronger deubiquitylating activity toward substrates that are also SUMOylated, providing a possible

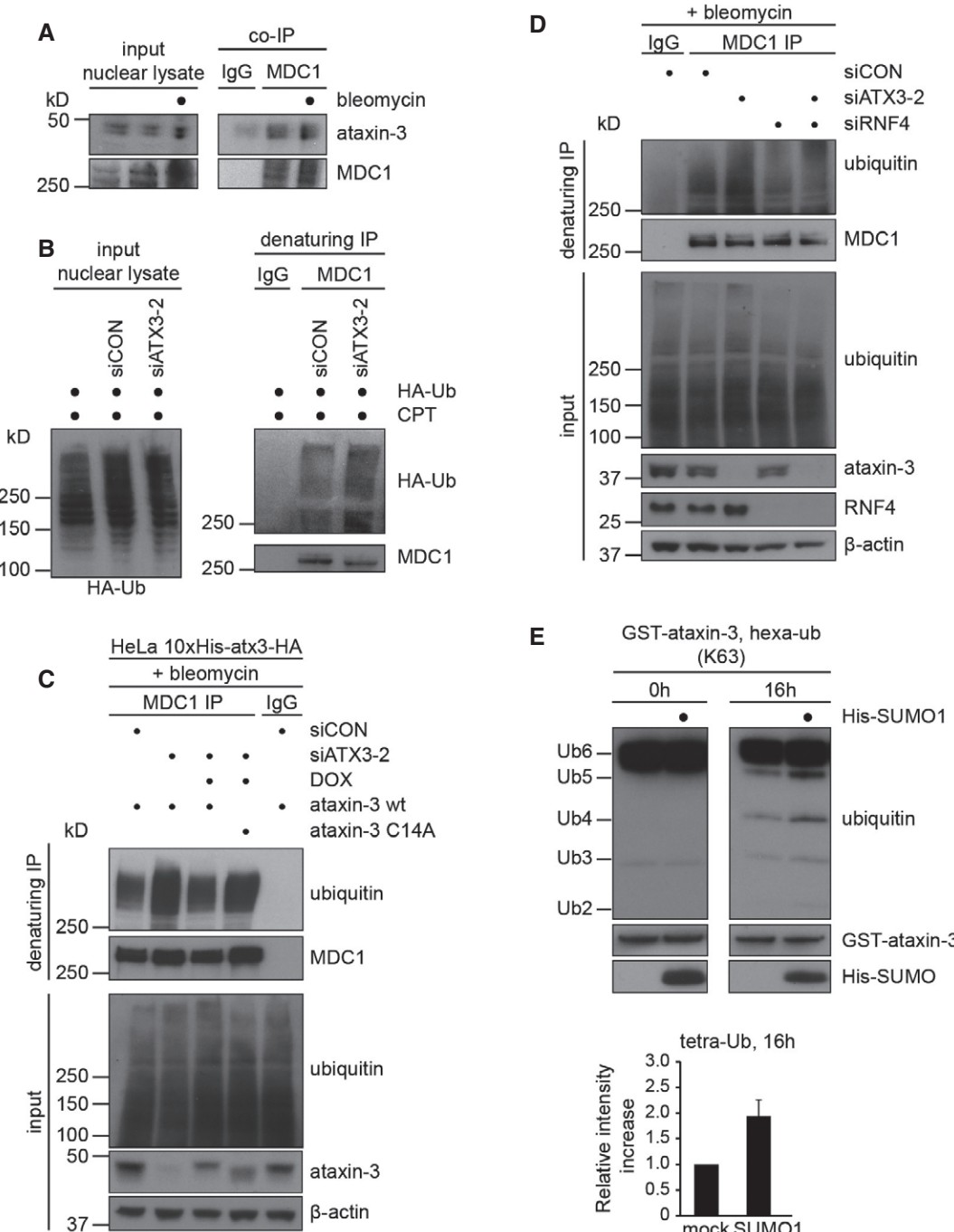

**Figure 5.  Ataxin-3 and RNF4 regulate ubiquitylation of MDC1.**

A   MDC1 interacts with ataxin-3. A co-immunoprecipitation of endogenous MDC1 from U2OS cells was performed. DSBs were induced or not by 10 μg/ml bleomycin, and MDC1 (or IgG) was immunoprecipitated from nuclear lysates followed by Western blotting with indicated antibodies.

B   U2OS cells were transfected with control or ataxin-3 siRNA (48 h) and HA-tagged ubiquitin (24 h). After induction of DNA damage with 20 μM camptothecin (CPT; 1 h), cells were fractionated in cytosolic and nuclear fractions and nuclei were re-suspended in denaturing buffer. MDC1 (or IgG) was immunoprecipitated from nuclear lysates followed by Western blotting with indicated antibodies.

C   Ubiquitylation assay of MDC1. HeLa cells expressing 10xHis-ataxin-3-HA wild-type or C14A cells were transfected with control or ataxin-3 siRNA (48 h) and were induced by 1 μg/ml Dox (24 h) where indicated. After the induction of DNA damage, cells were re-suspended in denaturing buffer. MDC1 (or IgG) was immunoprecipitated from lysates followed by Western blotting with indicated antibodies.

D   As in (B) but U2OS cells were transfected with ataxin-3 or RNF4 targeting siRNAs.

E   The catalytic activity of GST-ataxin-3 was analyzed *in vitro* using K63-linked hexa-ubiquitin as a substrate. Reactions were stopped at 0 or 16 h. Where indicated, GST-ataxin-3 was pre-incubated with recombinant His-SUMO1 before addition of the ubiquitin substrate. The relative intensity of tetra-ubiquitin product was quantified. Data are presented as mean ± SEM from three independent experiments.

molecular mechanism for its activity toward RNF4-ubiquitylated MDC1.

### Ataxin-3 positively regulates DSB-induced ubiquitylation

Our findings show that ataxin-3 affects ubiquitylation and the exchange kinetics of MDC1. Upon its recruitment to DSBs, MDC1 initiates an enzymatic cascade involving the association of the ubiquitin ligase RNF8 with phosphorylated MDC1 (Huen *et al*, 2007; Kolas *et al*, 2007; Mailand *et al*, 2007), which is followed by the association of RNF168 with RNF8-catalyzed ubiquitin conjugates on histone H1 (Thorslund *et al*, 2015). These events culminate in the conjugation of ubiquitin to lysine K13/15 of histone H2A (H2AK13/15Ub) (Fradet-Turcotte *et al*, 2013; Mattiroli *et al*, 2014), which promotes the ubiquitin-dependent recruitment of 53BP1 (Fradet-Turcotte *et al*, 2013) and the BRCA1 complex (Sobhian *et al*, 2007; Wang *et al*, 2007). To explore whether ataxin-3 affects DNA damage-induced signaling downstream of MDC1, we depleted ataxin-3 with two ataxin-3-specific siRNAs. An unbiased, semi-automated quantitative analysis (Appendix Fig S5) revealed that knock-down of ataxin-3 indeed impaired the accumulation of RNF8 (Fig 6A), RNF168 (Fig 6B), and ubiquitin conjugates at laser-induced DSBs (Fig 6C). Knock-down of ataxin-3 did not affect the steady-state levels of RNF8 (Appendix Fig S3B). We observed a reduction in RNF168 levels with one of the siRNAs directed against ataxin-3, whereas the second siRNA had no appreciable effect on the steady-state levels of RNF168 (Appendix Fig S3B). However, both siRNAs showed a similar effect on RNF168 recruitment, suggesting that the actual recruitment is impaired. Consistent with defective DSB-induced ubiquitylation, we found that depletion of ataxin-3 also significantly reduced both BRCA1 (Fig 6D) and 53BP1 accumulation (Fig 6E) to laser-inflicted DNA damage, as well as to ionizing radiation-induced foci (Appendix Fig S6A and B). Importantly, the recruitment of 53BP1 to laser-induced DNA damage was partially rescued in ataxin-3-depleted cells by stable expression of GFP-tagged wild-type ataxin-3, but not in cells expressing catalytically inactive GFP-ataxin-3[C14A] (Fig 6F and Appendix Fig S7A). We noted that the ectopic expression of GFP-ataxin-3 in cells depleted of endogenous ataxin-3 did not fully restore 53BP1 recruitment. We wondered whether this could be caused by the fact that the levels of the ectopically expressed GFP-ataxin-3 were higher than those observed for endogenous ataxin-3 (Appendix Fig S7B). Indeed, we found that over-expression of ataxin-3 in control cells already significantly reduced accrual of 53BP1 (Appendix Fig S7C). Therefore, the higher expression level of GFP-ataxin-3 compared to the endogenous expression level may be a plausible explanation for the observed partial rescue in our experimental set-up. These findings show that ataxin-3 by means of its deubiquitylating activity stimulates the ubiquitylation-dependent recruitment of BRCA1 and 53BP1 to sites of DNA damage.

To further strengthen the functional link between ataxin-3 and RNF4, we asked whether the antagonistic action between these enzymes observed on MDC1 (see Fig 5C) could also be observed downstream in the RNF8/RNF168 pathway. For this analysis, we focused on the recruitment of RNF168 itself, because the DSB-induced ubiquitylation of H2A, which triggers 53BP1 recruitment, is catalyzed by this enzyme (Mattiroli *et al*, 2012). We found that reducing the levels of RNF4 also resulted in a moderate decrease

in RNF168 recruitment although not to the same extent as caused by ataxin-3 depletion (Fig 6G and H). Interestingly, co-depletion of RNF4 and ataxin-3 did not further impair RNF168 recruitment, but instead restored RNF168 accumulation to a level that was almost comparable to the effect observed upon knock-down of RNF4 alone (Fig 6G and H, and Appendix Fig S8). Thus, RNF4 knock-down partially restored RNF168 recruitment in ataxin-3-depleted cells, consistent with a model in which the DUB activity of ataxin-3 opposes the ubiquitin ligase activity of RNF4 at DNA breaks.

### Depletion of ataxin-3 compromises DSB repair

Our findings suggest that ataxin-3 acts as an antagonist of RNF4 and regulates the initiation of the RNF8 pathway at the level of MDC1. Since RNF4 and MDC1 have been implicated in DSB repair by homologous recombination (HR) (Zhang *et al*, 2005; Huang *et al*, 2009; Lu *et al*, 2012; Luo *et al*, 2012) and non-homologous end joining (NHEJ) (Lou *et al*, 2006; Feng & Chen, 2012; Galanty *et al*, 2012), we asked whether ataxin-3 regulates these DSB repair mechanisms. To address a possible role of ataxin-3 in HR, we used cells with a stably integrated DR-GFP reporter, which relies on HR-dependent DSB repair for the restoration of GFP fluorescence following DSB induction by the I-SceI endonuclease (Pierce *et al*, 1999). Flow cytometric analysis revealed that HR, as measured by restoration of GFP fluorescence, was significantly inhibited by ataxin-3 depletion although not to the same extent as seen upon knock-down of the core HR factor BRCA2 (Fig 7A). A key feature of HR is that the ends of a DSB that arises in S/G2 phase are resected to generate single-stranded DNA overhangs, which are bound by RPA. In turn, RPA is displaced by RAD51, which plays a critical role in the search for and invasion of an undamaged DNA template (Roy *et al*, 2011). Of note, all these steps are known to be regulated by RNF4 (Galanty *et al*, 2012; Luo *et al*, 2012; Yin *et al*, 2012). To investigate the impact of ataxin-3 on these steps, we monitored the recruitment of RPA and RAD51 to DSBs in cells expressing fluorescently tagged geminin, which is an S/G2-specific cell cycle marker. We observed a significant reduction in RPA recruitment in S/G2 cells following laser micro-irradiation, consistent with a defect in DNA-end resection (Fig 7B). Impairment of HR was also supported by the observation that ataxin-3 depletion caused a decrease in the formation of RAD51 foci in S/G2 cells following exposure to IR (Fig 7C). Defects in HR are typically synthetic lethal in combination with PARP inhibition (McCabe *et al*, 2006). Indeed, cells in which ataxin-3 had been depleted became highly sensitive to PARP inhibitor (KU0058948) in clonogenic survival assays, an effect that was nearly similar to that seen after BRCA2 depletion (Fig 7D). These findings reveal that ataxin-3 function is important for efficient DSB repair by HR.

We subsequently investigated a possible role of ataxin-3 in NHEJ by using cells with a stably integrated EJ5-GFP reporter, which relies on NHEJ for the restoration of GFP fluorescence following DSB induction by the I-SceI endonuclease (Bennardo *et al*, 2008). Flow cytometric analysis revealed that the efficiency of NHEJ was severely compromised in ataxin-3-depleted cells to a similar extent as observed upon depletion of MDC1 (Fig 8A). To corroborate these findings, we used random plasmid integration into genomic DNA as a measure for NHEJ (Luijsterburg *et al*, 2016). Knock-down of the

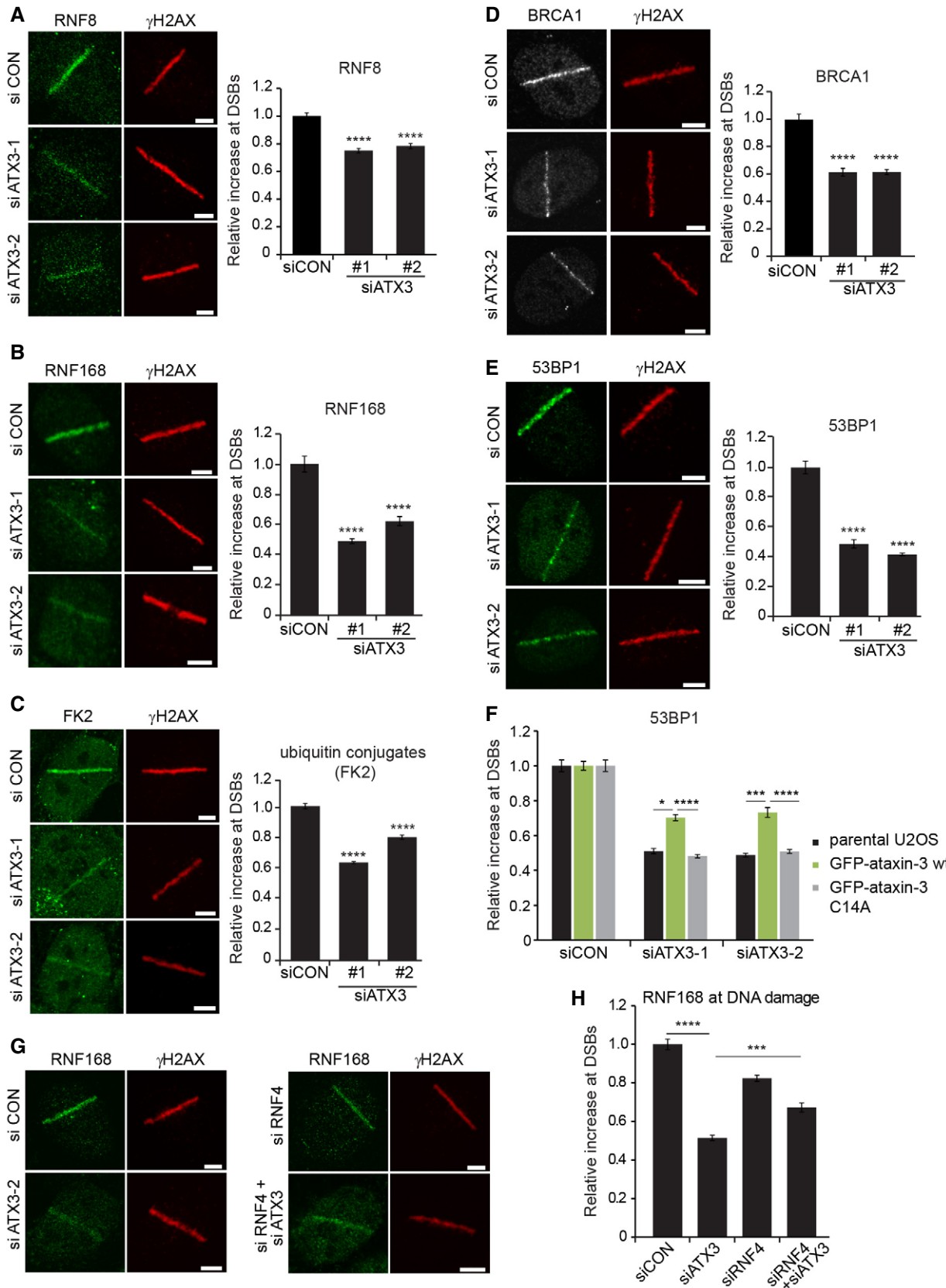

**Figure 6.**

**Figure 6.   Ataxin-3 is required for efficient DNA damage-induced ubiquitylation.**

A   U2OS cells were depleted of ataxin-3, laser micro-irradiated, and fixed after 15 min. Cells were immunostained for RNF8. Data are presented as mean ± SEM from two independent experiments. ****$P \leq 0.0001$ (Kruskal–Wallis test).

B   As in (A) but cells were analyzed for RNF168 accumulation at DSBs. Data are presented as mean ± SEM from three independent experiments. ****$P \leq 0.0001$ (Kruskal–Wallis test).

C   As in (A) but cells were analyzed for conjugated ubiquitin (FK2 antibody) accumulation at DSBs. Data are presented as mean ± SEM from two independent experiments. ****$P \leq 0.0001$ (Kruskal–Wallis test).

D   U2OS cells were transfected with control or two different ataxin-3 siRNAs, fixed at 1 h after laser micro-irradiation, and immunostained for BRCA1. Data are presented as mean ± SEM from three independent experiments. ****$P \leq 0.0001$ (Kruskal–Wallis test).

E   As in (D) but cells were immunostained for 53BP1. Data are presented as mean ± SEM from three independent experiments. ****$P \leq 0.0001$ (Kruskal–Wallis test).

F   Ataxin-3 was depleted in parental U2OS cells or in U2OS cells stably expressing wild-type GFP-ataxin-3 or catalytically inactive GFP-ataxin-3$^{C14A}$. Cells were subjected to laser micro-irradiation, and immunostained for 53BP1. Knock-down efficiencies of endogenous or ectopical GFP-tagged ataxin-3 were analyzed by immunoblotting, shown in Appendix Fig S7. Data are presented as mean ± SEM from three independent experiments. *$P \leq 0.05$; ***$P \leq 0.001$; ****$P \leq 0.0001$ (one-way ANOVA test).

G   U2OS cells depleted of ataxin-3, RNF4, or both or treated with control siRNA were laser micro-irradiated and fixed after 15 min. Cells were immunostained for endogenous RNF168. Depletion efficiencies of ataxin-3, RNF4, or both simultaneously were analyzed by immunoblotting, shown in Appendix Fig S8.

H   Quantification of the relative increase in RNF168 at DSBs (G). Data are presented as mean ± SEM from four independent experiments. ***$P \leq 0.001$; ****$P \leq 0.0001$ (Kruskal–Wallis test).

Data information: Scale bars, 5 μm.

core NHEJ factor KU80 almost completely abolished plasmid integration, showing that this process relies on NHEJ (Fig 8B). In line with the EJ5-GFP reporter data, knock-down of either MDC1 or ataxin-3 severely compromised NHEJ activity (Fig 8B). To further study the role of ataxin-3 and MDC1 in this repair pathway, we monitored the recruitment of the core NHEJ factor XRCC4 following laser micro-irradiation. While control cells showed clear accrual of endogenous XRCC4, depletion of ataxin-3 or MDC1 significantly impaired the recruitment of this NHEJ factor to laser-inflicted damage (Fig 8C). Importantly, ectopic expression of wild-type ataxin-3 partially restored XRCC4 recruitment in cells treated with ataxin-3-specific siRNAs, while catalytically inactive ataxin-3 failed to do so (Fig 8D), consistent with a role for the deubiquitylating activity of ataxin-3 in this process.

Finally, we addressed whether the functional antagonism between RNF4 and ataxin-3, which we observed to act on MDC1 exchange kinetics (see Fig 5C) and RNF168 recruitment (see Fig 6G and H), could also be observed during NHEJ in double knock-down experiments. In contrast to the strong impact of ataxin-3 knock-down, the depletion of RNF4 only had a mild impact on XRCC4 recruitment (Appendix Fig S9A). Interestingly, co-depletion of ataxin-3 and RNF4 partly restored XRCC4 recruitment to DSBs to the milder phenotype observed for the single depletion of RNF4 (Fig 8E). These experiments reveal an important role of ataxin-3 in regulating DSB repair by NHEJ and suggest that ataxin-3 regulates this repair pathway by antagonizing RNF4 activity. Considering the strong impact of ataxin-3 on both NHEJ and HR, we asked whether ataxin-3 is required for cells to cope with DSB-inducing agents. To this end, we performed clonogenic survival assays, which showed that ataxin-3 depletion renders cells hypersensitive to ionizing radiation (IR)-induced DSBs although not to the same extent as knock-down of XRCC4 (Appendix Fig S9B). The depletion of RNF4 had a more severe inhibitory effect on cell viability in response to IR compared with ataxin-3 depletion (Fig 8F), which may be due to its general role in controlling proteotoxic (Lallemand-Breitenbach *et al*, 2008; Tatham *et al*, 2008) and genotoxic stress (Prudden *et al*, 2007; Sun *et al*, 2007). Importantly, co-depletion of RNF4 and ataxin-3 did not further compromise cell viability but instead improved the ability of the cells to survive upon exposure to IR, again supporting antagonistic activities of ataxin-3 and RNF4

(Fig 8F). Notably, co-depletion of RNF4 did not fully restore the sensitivity of ataxin-3-depleted cells to IR, suggesting that ataxin-3 may play additional roles in the cellular response to DSBs. Together our data suggest that the opposing activities of RNF4 and ataxin-3 regulate the signaling and repair of DSBs.

# Discussion

In this study, we identified the DUB ataxin-3 as an important player in the cellular response to DSBs. We uncovered that ataxin-3 localizes to DNA lesions directly after their occurrence in a fashion that requires PIAS4/Ubc9-dependent SUMOylation. Our study reveals a novel regulatory mechanism in which ataxin-3 opposes the activity of the STUbL RNF4, which ubiquitylates SUMO-modified proteins such as MDC1 (Praefcke *et al*, 2012). In the case of MDC1, RNF4-mediated ubiquitylation facilitates the removal of SUMOylated MDC1 from DNA damage (Galanty *et al*, 2012; Luo *et al*, 2012; Yin *et al*, 2012). We propose that the immediate presence of ataxin-3 at DNA lesions prevents premature removal of MDC1 from DSBs and by this means reinforces initiation of DNA damage signaling and DNA repair (Fig 9).

Our study revealed that depletion of ataxin-3 increases the levels of ubiquitylated MDC1, while at the same time reducing its residence time at DNA breaks, consistent with the idea that ubiquitylation of MDC1 is the primary signal that regulates its release from chromatin. Additionally, our findings reveal that ubiquitylation of MDC1 is regulated by ataxin-3 in a manner that requires its catalytic activity. Notably, whereas ataxin-3 depletion had a strong effect on the MDC1 exchange at sites of DNA damage, the net effect on the steady-state accumulation of MDC1 at DSBs was subtle, similar to what has been found for RNF4 depletion (Galanty *et al*, 2012). This suggests that the rapid dissociation of MDC1 in the absence of ataxin-3 is likely to be compensated by the instant recruitment of new MDC1 molecules, resulting in a dynamic cycle in which the residence time of individual molecules is severely reduced and insufficient to properly activate the response to DSBs. Consistent with this model, we found that all events downstream of MDC1, such as DSB-induced chromatin ubiquitylation and the subsequent ubiquitin-dependent recruitment of BRCA1 and 53BP1, are impaired

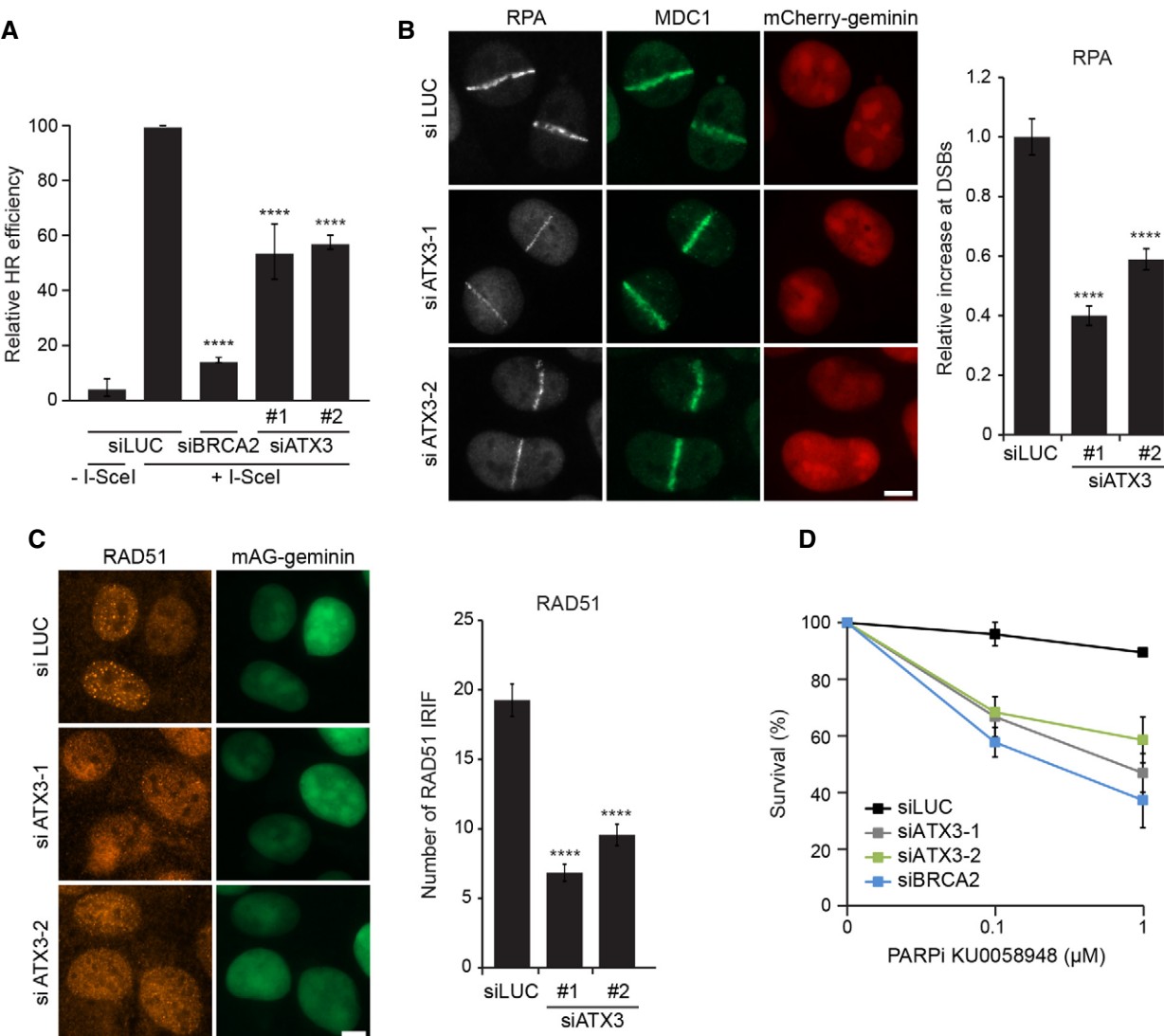

**Figure 7.   Homologous recombination is less efficient in the absence of ataxin-3.**

A   HEK293 cells containing a stably integrated copy of the DR-GFP reporter for HR were depleted of ataxin-3, BRCA2 (positive control), or luciferase (negative control). The repair of I-SceI-induced DSBs by HR was quantified. Data are presented as mean ± SEM from two independent experiments. ****$P \leq 0.0001$ (one-way ANOVA test).

B   mAG-geminin-expressing U2OS cells were transfected with indicated siRNAs, subjected to laser micro-irradiation, and immunostained for RPA and MDC1. The intensity of RPA accumulated at sites of laser damage was quantified. Data are presented as mean ± SEM from three independent experiments. ****$P \leq 0.0001$ (Kruskal–Wallis test).

C   U2OS cells expressing mAG-geminin were depleted of ataxin-3, exposed to ionizing radiation, and immunostained for RAD51. The number of RAD51 foci was quantified. Data are presented as mean ± SEM from three independent experiments. ****$P \leq 0.0001$ (Kruskal–Wallis test).

D   VH10-SV40 cells were transfected with indicated siRNAs, seeded at low density, and exposed for 7 days to different concentrations of PARPi (KU0058948). Colonies of more than 10 cells were scored. Data are presented as mean ± SEM from two independent experiments.

Data information: Scale bars, 5 μm.

in cells depleted of ataxin-3. Although we cannot exclude that other ataxin-3-dependent mechanisms contribute to these events, our results suggest that these downstream defects in ataxin-3-depleted cells might originate from the inability to efficiently retain MDC1 at DSBs.

In earlier studies, it has been proposed that the timely RNF4-mediated extraction of MDC1, once the response to DSBs has been initiated, is important to allow access of downstream signaling and

repair proteins to DSBs (Galanty *et al*, 2012; Luo *et al*, 2012; Yin *et al*, 2012). It is noteworthy, however, that our data, as well as an earlier study (Vyas *et al*, 2013) suggest that RNF4 accumulates almost instantly at DSBs at a time when MDC1 removal is expected to be counterproductive for activation of DNA damage signaling. We therefore propose a two-step mechanism with an early phase during which ataxin-3 counteracts RNF4-mediated MDC1 removal safeguarding activation of the DSB response and a late phase when

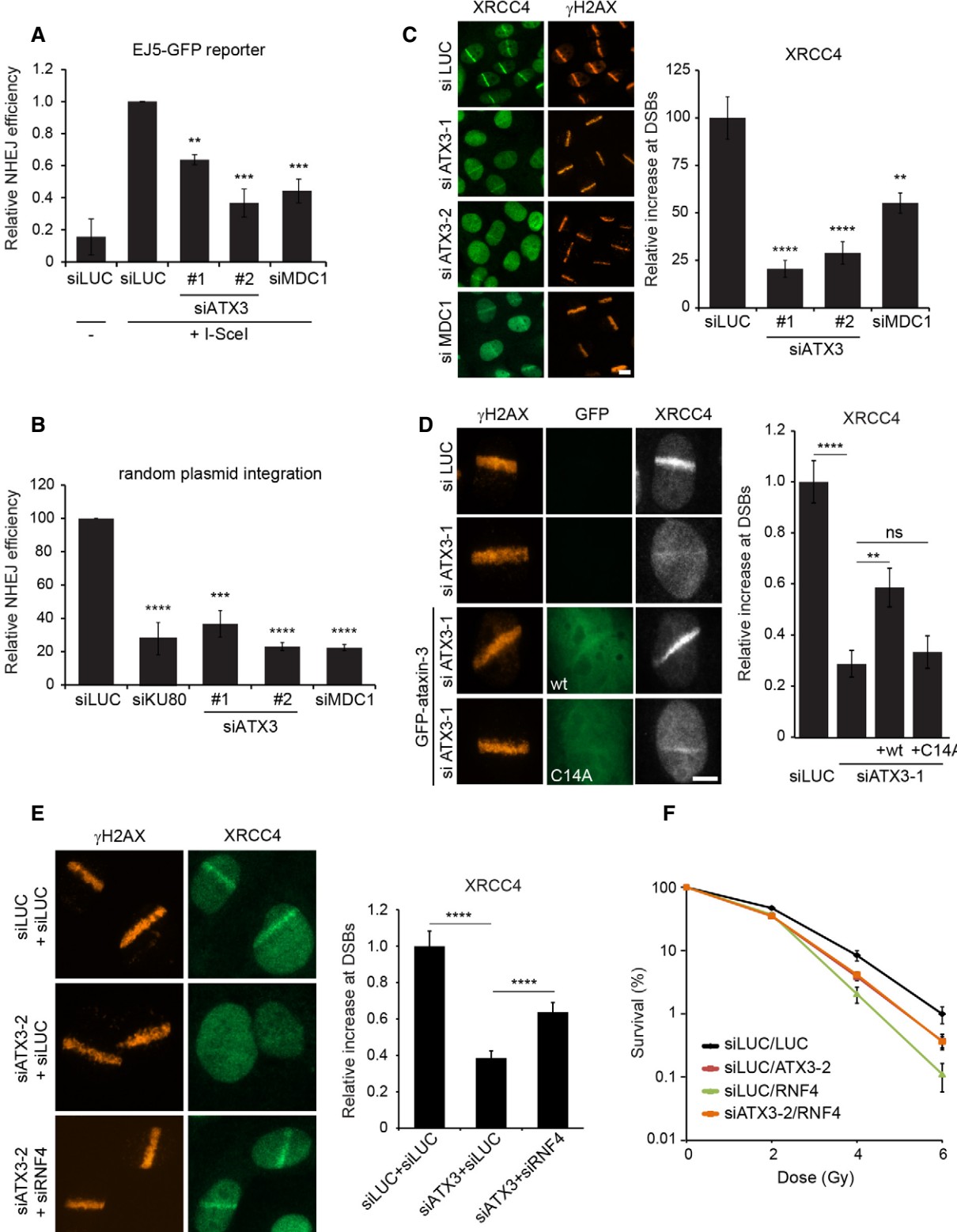

**Figure 8.**

RNF4 stimulates eviction of MDC1 thereby promoting the successive events in DNA damage signaling. It is tempting to speculate that the intrinsic mode of action of RNF4 to remove MDC1 from chromatin may prevent erroneous induction of the cellular response to DSBs as it requires the co-recruitment of ataxin-3 for proper activation. As such, ataxin-3 may provide another layer of safety checks that

**Figure 8.   Depletion of ataxin-3 compromises NHEJ.**

A   HEK293 cells containing a stably integrated copy of the EJ5-GFP reporter for NHEJ were depleted of ataxin-3 or MDC1 and repair of I-SceI-induced DSBs by NHEJ was quantified. Data are presented as mean ± SEM from three independent experiments. **$P \leq 0.01$; ***$P \leq 0.001$ (one-way ANOVA test).

B   Random plasmid integration assay. U2OS cells were transfected with indicated siRNAs and with linearized pEGFP-C1 plasmid DNA. After 5 days, cells were counted and re-seeded in medium containing or lacking G418. Colonies were counted on day 15. Data are presented as mean ± SEM from two independent experiments. ***$P \leq 0.001$; ****$P \leq 0.0001$ (one-way ANOVA test).

C   U2OS cells were depleted of ataxin-3 or MDC1, subjected to laser micro-irradiation and immunostained for XRCC4 and γH2AX. The relative increase in XRCC4 at laser-induced DNA damage was quantified. Data are presented as mean ± SEM from two independent experiments. **$P \leq 0.01$; ****$P \leq 0.0001$ (Kruskal–Wallis test).

D   Parental U2OS cells or U2OS cells stably expressing GFP-ataxin-3 or GFP-ataxin-3$^{C14A}$ were depleted of ataxin-3 (siATX3-1). After micro-irradiation, cells were immunostained for XRCC4. The levels of XRCC4 at sites of laser-induced DNA damage were quantified. Data are presented as mean ± SEM from two independent experiments. **$P \leq 0.01$; ****$P \leq 0.0001$ (Kruskal–Wallis test). ns, non-significant.

E   As in (C) but cells were transfected with siRNAs targeting ataxin-3 or RNF4. Data are presented as mean ± SEM from two independent experiments. ****$P \leq 0.0001$ (Kruskal–Wallis test).

F   For survival experiments, VH10-SV40 cells were transfected with indicated siRNAs, seeded at low density and exposed to the indicated doses of ionizing radiation. Cells were incubated for 7 days and stained with methylene blue. Colonies of more than 10 cells were scored. Data are presented as mean ± SEM from two independent experiments.

Data information: Scale bars, 5 μm.

protects cells from superfluous activation of the DSB response by chromatin-localized MDC1.

How is the activity of ataxin-3 toward MDC1 regulated? Previous studies have shown that MDC1 is modified with SUMO and ubiquitin in response to DNA damage (Galanty *et al*, 2012; Luo *et al*, 2012; Yin *et al*, 2012), which is regulated through an interaction between the SIM domain in RNF4 and SUMO-modified MDC1 (Luo *et al*, 2012; Yin *et al*, 2012). In contrast, we can detect a constitutive interaction between MDC1 and ataxin-3 that is neither enhanced by DNA damage, nor dependent on the primary SUMOylation site (K1840) in MDC1 (Luo *et al*, 2012). Additionally, our findings reveal that ataxin-3 can directly interact with SUMO1 and that this interaction stimulates the deubiquitylating activity of the enzyme. Thus, rather than the MDC1-ataxin-3 interaction, our results suggest that the activity of ataxin-3 toward MDC1 is regulated by DNA damage-induced SUMOylation. Whether the activity of ataxin-3 toward ubiquitylated MDC1 is triggered by the SUMOylation of MDC1 itself or another DNA damage-associated protein remains to be established. Even though ataxin-3 sequestration does not require SUMOylated MDC1, our data unequivocally show that the recruitment of ataxin-3 to DSBs is strictly dependent on SUMOylation. While the nature of the SUMO modification that triggers chromatin accrual of ataxin-3 remains to be resolved, we consider it likely that not a single substrate, but the collective DNA damage-induced SUMOylation of chromatin-associated proteins may facilitate this process.

The role of ataxin-3 in counteracting RNF4 and regulating the chromatin retention of MDC1 may also, in part, explain its requirement for efficient DSB repair by both NHEJ and HR. Previous studies have shown that loss of RNF4 also leads to defects in NHEJ (Galanty *et al*, 2012) and HR (Luo *et al*, 2012). Moreover, overexpression of the BRCT domain of MDC1 was shown to confer a defect in NHEJ as well, implicating a role for full-length MDC1 in this repair pathway (Stucki *et al*, 2005). Additionally, knock-down of MDC1 was shown to impair DSB repair by both NHEJ and HR (Luo *et al*, 2012; Zhang *et al*, 2005). Here, we confirm such a role by demonstrating that knock-down of MDC1 leads to a pronounced defect in NHEJ. Loss of MDC1 function and depletion of ataxin-3 both result in a failure to recruit RNF168 to chromatin surrounding DSBs ultimately culminating in the defective recruitment of 53BP1 (Kolas *et al*, 2007). It is noteworthy that defects in either RNF168

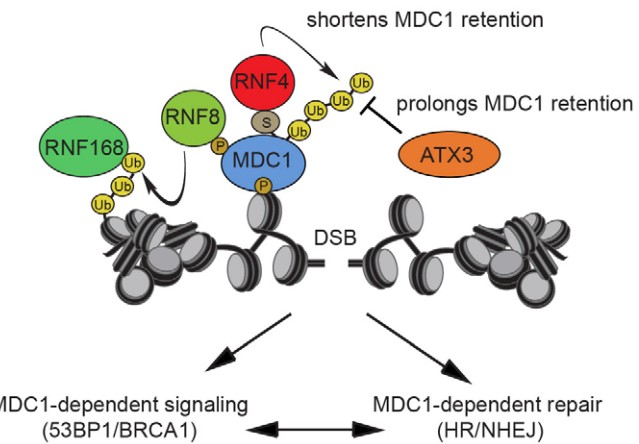

**Figure 9.   Model of the regulatory role of ataxin-3 in consolidating the MDC1-dependent DNA double-strand break response.**
During the early phase of the response to DSBs, ataxin-3 is counteracting the RNF4-induced chromatin removal of MDC1. By prolonging the residence time of MDC1 at chromatin, ataxin-3 ensures that the damage response is accurately activated.

or 53BP1 in knock-out mice and human patients are associated with class-switch recombination defects in B cells caused by defective DSB repair by NHEJ (Stewart *et al*, 2007; Li *et al*, 2010; Ramachandran *et al*, 2010). Similarly, MDC1-deficient mice also suffer from immunodeficiency and display moderately decreased levels of class-switch recombination (Lou *et al*, 2006), consistent with the NHEJ defect we observe in our assays. These findings suggest that the opposing activities of ataxin-3 and RNF4 that act upon MDC1 are important to ensure efficient DSB repair by NHEJ and HR. In this light, it is noteworthy that also recruitment of RPA, which is subject to DNA damage-induced SUMOylation (Dou *et al*, 2010) and regulated by RNF4 (Galanty *et al*, 2012; Yin *et al*, 2012), is reduced in ataxin-3-depleted cells, suggesting that RPA may also be subject to the counteracting activities of RNF4 and ataxin-3.

It is not uncommon that ubiquitin ligases and DUBs operate in functional pairs that counteract or fine-tune each other's activities toward substrates (Komander *et al*, 2009). Our findings reveal a

reoccurring antagonism between RNF4 and ataxin-3 in regulating different aspects of the response to DSBs. In particular, we suggest that ataxin-3 is a functional antagonist of RNF4 during the signaling and repair of DSBs where it acts as a SUMO-activated DUB on at least one of the identified RNF4 substrates: MDC1. Whether ataxin-3 also acts on other RNF4 substrates during the DDR remains to be established. A number of studies have revealed that, in addition to DNA lesions, there are other nuclear locations where groups of proteins are SUMOylated such as PML bodies, telomeres, and nucleoli (Jentsch & Psakhye, 2013) and it is feasible that the concerted action of RNF4 and ataxin-3 also plays regulatory roles at these SUMOylation hubs. Moreover, RNF4 and ataxin-3 have also both been implicated in protein quality control and are part of the cellular defense system that copes with conditions of proteotoxic stress (Warrick *et al*, 2005; Guo *et al*, 2014). It therefore remains possible that ataxin-3 also acts as a SUMO-activated DUB in protein quality control, which may be of relevance for its protective activity against neurodegeneration-associated proteins (Warrick *et al*, 2005).

# Materials and Methods

Supplementary Materials and Methods can be found in the Appendix.

### UV-A laser micro-irradiation

Micro-irradiation was performed as previously reported (Acs *et al*, 2011). Living cells were examined in Leibovitz's L-15 medium (Life Technologies) supplemented with 10% FBS. Cells were fixed at indicated time points after micro-irradiation. The relative accumulation of proteins at laser-inflicted DNA damage lines was determined using an unbiased, semi-automated quantification approach using the Volocity software (PerkinElmer). In brief, microscopy images of γH2AX immunostaining and Hoechst staining were used for segmentation of the DNA damage and the nucleus area, respectively. Fluorescence intensities in these segmentations were measured, and a relative enrichment of proteins of interest at DSBs was calculated by taking a ratio of the mean intensity in the DNA damage area versus the mean intensity in the non-DNA damage area.

### Generation of DSBs

Ionizing radiation was delivered by a YXlon X-ray generator (YXlon International, 200 KV, 10 mA, dose rate 2 Gy/min).

### IRIF analysis

IRIF were evaluated in ImageJ, using a custom-built macro that enabled automatic and objective analysis of the foci (Typas *et al*, 2015). In brief, cell nuclei were detected by thresholding the (median-filtered) DAPI signal, after which touching nuclei were separated by a watershed operation. The foci signal was background-subtracted using a difference of Gaussians filter. For every nucleus, foci were identified as regions of adjacent pixels satisfying the following criteria: (i) The gray value exceeds the nuclear background signal by a set number of times (typically 2–4×) the median background standard deviation of all nuclei in the image and is higher than a user-defined absolute minimum value; (ii) the area is larger than a defined area (typically 2 pixels). These parameters were optimized for every experiment by manually comparing the detected foci with the original signal.

### Exchange kinetics of GFP-MDC1

U2OS cells stably expressing GFP-MDC1 were grown on 18-mm coverslips. Cells were placed in a Chamlide CMB magnetic chamber, and the growth medium was replaced by $CO_2$-independent Leibovitz's L-15 medium supplemented with 10% FCS and penicillin–streptomycin. Laser micro-irradiation was carried out on a Leica SP5 confocal microscope equipped with an environmental chamber set to 37°C. DSB-containing tracks (1.0 μm width) were generated with a Mira mode-locked titanium-sapphire (Ti: Sapphire) laser ($\lambda$ = 800 nm, pulse length = 200 fs, repetition rate = 76 MHz, output power = 80 mW) using a UV-transmitting 63 × 1.4 NA oil immersion objective (HCX PL APO; Leica) at zoom 2.5 (scan speed 400 Hz, line average 2). GFP-MDC1 was allowed to accumulate at laser-induced DNA damage for 3 min. Half of the laser tracks (10 × 2 μm) was bleached using 2 consecutive bleach pulses with the argon lines 488 nm + 514 nm at zoom 6 (scan speed 400 Hz, line average 2). The loss of fluorescence (FLIP) in the non-bleached half and the recovery of fluorescence (FRAP) in the bleached half were monitored with low laser power for ~6 min using the 488 laser line. The exchange rate of GFP-MDC1 is represented as FLIP-FRAP (normalized to 1).

### HR and NHEJ reporter assays

HEK293 cells containing a stably integrated copy of either the DR-GFP or EJ5-GFP reporter were used to measure the repair of I-SceI-induced DSBs by HR or NHEJ, respectively (Pierce *et al*, 1999; Bennardo *et al*, 2008). Briefly, 48 h after siRNA transfection, cells were transfected with I-SceI and mCherry expression vectors. FACS analysis was used 48 or 72 h after plasmid transfection to measure the fraction of GFP-positive cells among the mCherry-positive cells on a LSRII flow cytometer (BD Bioscience) using FACSDiva software version 5.0.3. Quantifications were performed using Flowing Software (www.flowingsoftware.com).

### Random plasmid integration assay

U2OS cells were seeded (day 1) and transfected with siRNAs in a 6-cm dish the following day (day 2). Later that day, the cells were transfected with 2 μg gel-purified *Bam*HI-*Eco*RI-linearized pEGFP-C1 plasmid. The cells were subsequently transfected twice with siRNAs at 24 and 36 h after the first transfection (day 3 and day 4, respectively). On day 5, cells were collected, counted, and seeded in 15-cm dishes either lacking or containing 0.5 mg/ml G418. The transfection efficiency was determined on the same day by FACS analysis. The cells were incubated at 37°C to allow colony formation, and medium was refreshed on day 8 and 12. On day 15, the cells were washed with 0.9% NaCl and stained with methylene blue. Colonies of more than 50 cells were scored. Random plasmid integration events on the G418-containing plates were normalized to the plating efficiency (plate without G418) and transfection efficiency (based on GFP expression).

## SUMO binding assay from cell lysates

Cells were pre-treated or not with 10 μg/ml bleomycin for 1 h to induce DNA damage before cells were harvested and washed twice in ice-cold 1× PBS (500 *g*, 3 min at 4°C). Lysis was performed on ice for 30 min in EBC buffer (50 mM Tris pH 7.5, 150 mM NaCl, 1 mM EDTA, 1 mM DTT, 1× protease inhibitor cocktail, 20 mM NEM, 1 mM Na$_3$VO$_4$, 1 mM β-glycerophosphate, 10 μM MG132) supplemented with 0.5% NP-40 and 250 U benzonase. After clearing (16,000 *g*, 10 min at 4°C), the lysates were incubated with 20 μl of empty agarose beads (negative control) or recombinant SUMO1 or SUMO2 immobilized on agarose beads (Enzo Life Sciences) for 2 h at 4°C under constant rotation. The beads were washed five times with EBC buffer and eluted with 2× LDS NuPAGE sample buffer containing reducing agent (Life Technologies).

## *In vitro* SUMO binding assay

Unconjugated agarose beads (negative control) or recombinant SUMO1 immobilized on agarose beads (Enzo Life Sciences) were washed three times in buffer A (50 mM Tris pH 7.5, 150 mM NaCl, 1 mM DTT, 0.5% NP-40) and collected at 2,500 *g* 2 min 4°C. One microgram recombinant ataxin-3 was added to the beads and incubated for 2 h at 4°C under constant rotation. Beads were washed six times with buffer A and eluted with 2× LDS sample buffer containing reducing agent. 50 ng recombinant ataxin-3 in 2× LDS sample buffer containing reducing agent was loaded as input.

## *In vitro* DUB assay

Ubiquitin-hydrolyzing activity of GST-ataxin-3 was measured *in vitro* on hexa-ubiquitin (K63-linked) chains (Boston Biochem). 0.5 μg recombinant GST-ataxin-3 was pre-incubated for 30 min at RT with 1 μg recombinant His-SUMO1 (Enzo Life Sciences) in DUB buffer (50 mM HEPES pH 8, 0.5 mM EDTA, 1 mM DTT, 0.1 μg/ml BSA). 0.25 μg hexa-ubiquitin (K63) substrate (diluted in 50 mM Tris pH 7.5) was added, and the mixture was incubated for 0 h or 16 h at 37°C. The reaction was stopped by addition of 2× LDS sample buffer containing reducing agent. Prior to SDS–PAGE analysis, samples were pre-heated at 37°C for 20 min.

**Expanded View** for this article is available online.

## Acknowledgements

We thank Drs. Georges Mer, Ron Hay, Roger Greenberg, Alfred Vertegaal, Marianne Farnebo, Jacques Neefjes, Henry Paulson, Michael Huen, Thorsten Schmidt, Thomas Helleday, Jorma Palvimo, Jeremy Stark, Anne Simonsen, Kristijan Ramadan, Maria Jasin, and Zhenkun Lou for reagents, help, and advice and the members of the Dantuma and Van Attikum laboratories for help and input. This work was supported by the Swedish Cancer Society (NPD), the Swedish Research Council (KA, NPD), Mayo Clinic-Karolinska Institute collaborative grant (NPD), the Karolinska Institute (NPD), NWO-VENI (MSL), FEBS Fellowship (MSL), and an ERC Consolidator grant (HvA).

## Author contributions

AP, MSL, KA, HvA, and NPD designed the research; AP, MSL, KA, WWW, AH, LKH, MM, and CB performed the experiments; AP, MSL, KA, AH, FAS, HvA, and NPD analyzed the data; and AP, MSL, HvA, and NPD wrote the manuscript.

## Conflict of interest

The authors declare that they have no conflict of interest.

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
