## [Review Process File · The EMBO Journal]

Manuscript EMBO-2016-95151

Ataxin-3 consolidates the MDC1-dependent DNA double-strand break response by counteracting the SUMO-targeted ubiquitin ligase RNF4

Annika Pfeiffer, Martijn S. Luijsterburg, Klara Acs, Wouter W. Wiegant, Angela Helfricht, Laura K. Herzog, Melania Minoia, Claudia Böttcher, Florian A. Salomons, Haico van Attikum and Nico P. Dantuma

*Corresponding authors: Haico van Attikum Leiden University Medical Center;
Nico P. Dantuma, Karolinska Institute*

Review timeline:

Submission date:	01 July 2016
Editorial Decision:	02 August 2016
Revision received:	29 December 2016
Editorial Decision:	11 January 2017
Revision received:	02 February 2017
Accepted:	06 February 2017

Editor: Hartmut Vodermaier

Transaction Report:

1st Editorial Decision

02 August 2016

Thank you again for submitting your manuscript on ataxin-3 as a SUMO-targeted DUB in the DSB response. We have now received comments from three expert referees, which I am copying below for your information. Based on these reports, we appreciate that there is potential interest, but it is also clear that all referees retain presently major reservations regarding the depth of understanding and the experimental support for some of the main conclusions. In particular, referees 1 and 2 ask for more conclusive insight into how ataxin-3 fits into our current picture of the DNA damage response, and referee 3 remains unconvinced that you have decisively demonstrated that ataxin-3 acts as a SUMO-targeted DUB. Given these substantial concerns and limitations, I am afraid we cannot consider the study a sufficiently compelling candidate for EMBO Journal publication, at least at the present stage.

Given the principle overall interest of this work and the very specific nature of the concerns being raised, I would nevertheless be willing to give you an opportunity to respond to the referees' criticisms by way of a revised version of the manuscript. Therefore, should you be able to validate and deepen the present analyses along the lines suggested by all referees, we would remain open to considering this study further for publication. Please note however that given this may require substantial further time and efforts, and that I am presently not able to predict the outcome of an eventual re-evaluation, which will depend on the completeness of your responses and the extent of further insight obtained. Since it is our policy to allow only a single round of major revision, I would

be happy to discuss a possible extension of our normal three-months revision period - during which time the publication of any competing work elsewhere would have no negative impact on our final assessment of your own study. Also, should you have any specific questions/comments regarding the referee reports or your revision work, please do not hesitate to get in touch with me ahead of time.

REFEREE REPORTS

Referee #1:

This is a very interesting report and the notion of a SUMO-directed DUB as far as I am aware is novel. It is also interesting to see an impact of SUMO modification so early in the damage response, an observation largely at odds with the RNF4-findings with which this report conceptually fits.

Overall I am supportive of publication of the manuscript however there is some confusion about how Atx3 fits into the DDR that needs clarification and one technical aspect that concerns me.

Technical concern.

Worryingly expression of WT-Atx3 doesn't fully rescue its depletion (~70% rescue. Figure 1C). This may be because over-expression is also deleterious to 53BP1 recruitment, or because the siRNAs hit another target. As the majority of the manuscript does not revisit complementation, but uses siRNA throughout there is a concern that some of the impacts seen are due to the non-specific knock-down. I would be reassured if the more vital findings were accompanied by complementation experiments (eg MDC1 turn-over, cell survival, NHEJ assays). (There is also no expression level to accompany Fig 1 C, although something similar is in fig 4C).

Ax3 in the DDR.

The model presented suggests an MDC1- SUMO-directed DUB regulates NHEJ, this needs more support.

If recruitment of Atx3 is PIAS4/Ubc9 dependent and also dependent on SIM3 of Atx3 then presumably interaction with MDC1 is also SUMO dependent. The immunoprecipitation (Fig 4A) therefore should show SUMO dependence. Could the authors repeat this experiment in the presence and absence of PIAS4 siRNA for example, or with the K1840R-MDC1 mutant - which is reported to be the main SUMO1-modification site on MDC1?

The relationship between ATx-3 and RNF4 (the presumed ligase it is countering) at the level of MDC1 would be expected to be key. Yet this has not been explored. Does reducing RNF4 levels reduce the levels of MDC1 ubiquitination in Atx3 depleted cells? Does loss of RNF4 restore normal early MDC1 turn-over-kinetics (as might be expected from the RNF168 data) what is the impact of RNF4 depletion on XRCC4? MDC1-K1840R would presumably be immune to both Atx3 and RNF4 -mediated regulations - confirming the role of this sites SUMO-modification as the likely route of regulation.

The link between increased MDC1 turn-over seen on Atx3 depletion at early time points and subsequent impact on DDR biology is difficult to see from the presented experiments. Why would the rapid re-association of MDC1 have a negative down-stream effect if other regulation is normal and MDC1 is not lost from lines of damage? Is the 'new' MDC1 not modified? What happens to RNF8 recruitment? How is the impact on RNF168 achieved - do the authors also think there is a non-MDC1 effect (say on H1-ubiquitination) ? Is this responsible for the NHEJ defect? How is the impact on XRCC4 achieved? Is this responsible for the NHEJ defect?

Functionally the model is not tied in. We might anticipate that RNF4 depletion in partially rescuing RNF168 (for whatever reason) may also restore some NHEJ and IR resistance of Atx3 depleted cells. Is this the case?

An unexplored and facile possibility is that DDR protein expression is affected. A simple blot showing MDC1, RNF8, RNF4 ,RNF168, 53BP1 protein levels is needed to negate this possibility.

Referee #2:

In this study, Pfeiffer and co-workers identified a deubiquitylation enzyme (DUB), ataxin3, counteracts RNF4 and regulates the retention of MDC1. Ataxin3 functions in a SUMO dependent way. They found that depletion of ataxin3 could decrease the chromatin-dwell time of MDC1 at the damage sites and impairs the recruitment of the ubiquitin ligase RNF168. Depletion of ataxin3 also impairs NHEJ and sensitizes cells to IR. The authors proposed that the counteraction between ataxin3 and RNF4 provides a mechanism that could tightly regulates the MDC1 dependent signaling and DNA damage response. These findings are interesting and are likely to be of interest for a broad audience. However, the study needs to be strengthened with more evidence.

Major concerns:

1. The authors showed that ataxin3 binds to MDC1 in Figure 4A. Is MDC1 required for ATX3 recruitment? Does the SIM of ATX3 bind to sumoylated MDC1?
2. In Figure 4, the authors should include cells rescued with WT/CA/*SIM mutants of ataxin3 in the experiment to confirm the deubiquitination of MDC1 is affected by ataxin3 and it depends on the SIMs.
3. Why RNF4 depletion reduced RNF168 accumulation?
4. As to the functional assay, the authors showed that depletion of ataxin3 affects the NHEJ. However, they did not show any data about HR. It would be better to check the effect on HR too. They should also check the recruitment of RNF8 and BRCA1.

Minor concerns:

1. In abstract, the authors claimed that "Depletion of ataxin-3 increases the levels of ubiquitylated MDC1, which coincides with an increase in the chromatin-dwell time of MDC1 at DSBs". This is opposite what is reported in the text.
2. Figure 1B, the authors should include a statistic analysis of the quantification.
3. Figure 1C, the authors should show a representative picture of the experiment.
4. In the text, from line 127 to line 141, all the Fig2 E,F,G are mis-referred.

Referee #3:

Pfeiffer et al. reported a deubiquitylation enzyme (DUB) ataxin-3 involved in DNA double strand break (DSB) response.

In this manuscript, authors showed that ataxin-3 was recruited to DSB induced by laser micro-irradiation or bleomycin treatment. The retention of Ataxin-3 to DBS was reported to depend on SUMO binding rather than its ubiquitin interaction motif (UIM) and DUB activity. They suggested Ataxin-3 enhanced the assembly of 53bp1 to DSB by regulating chromatin retention of MDC1, therefore supported NHEJ repair.

Much of the data presented to support the contentions above is marginal and not very convincing. A case in point being Fig 4 where the differences in the retention time on chromatin are small. While ataxin-3 may indeed be recruited to DSBs this is not really new, but is consistent with a previously published report (Nishi et al., 2014).

What is new here is the idea that ataxin-3 is recruited to DSBs by interaction with SUMO. This is based on knocking down SUMO conjugating enzymes and mutation of the putative SUMO interaction motif in ataxin-3. However a large number of proteins at DSBs are SUMO modified and the authors did not present evidence that the protein containing the SIM mutant was functional. The pull downs from cell extracts are not really informative as we have no idea of how direct the interact is.

What is missing from this paper is a direct demonstration that ataxin-3 is indeed a SUMO targeted

DUB. This requires an in vitro experiment with purified recombinant ataxin-3 and a variety of recombinant substrates that are both SUMO modified and ubiquitinated. Such substrates could include ubiquitinated SUMO chains that are the product of RNF4 action or a substrate that is modified at different sites by SUMO and ubiquitin. The defining feature of a SUMO targeted DUB is that DUB activity should be enhanced by SUMO modification and reduced by removal of SUMO. In fact this would be manifest in a reduced K_m for substrate in the presence of SUMO. Without such data the statement that ataxin-3 is a SUMO targeted deubiquitination enzyme cannot be justified.

1st Revision - authors' response

29 December 2016

Response to the referees

We would like to thank the reviewers for their insightful and constructive comments, which have been very helpful in improving the manuscript. The revision contains a large number of new data that strengthen the novel link between the deubiquitylating enzyme ataxin-3 and the SUMO-targeted ubiquitin ligase RNF4. Most importantly, we show now in various ways that ataxin-3 and RNF4 have opposing activities in the DNA-double-strand break (DSB) response: 1) regulation of MDC1 chromatin residence, 2) MDC1 ubiquitylation, 3) RNF168 recruitment, 4) XRCC4 recruitment and 5) cell survival after exposure to ionizing radiation. An unexpected but intriguing new finding is our observation that SUMO has a stimulatory effect on the catalytic activity of ataxin-3. To better emphasize the conceptual advances of this study, we decided to change the title of the manuscript and present the data in a different order with a focus on the link between ataxin-3 and RNF4. The parts in the manuscript that refer to new data or are made in direct response to the comments of the reviewers are marked in red font.

Referee #1

“This is a very interesting report and the notion of a SUMO-directed DUB as far as I am aware is novel. It is also interesting to see an impact of SUMO modification so early in the damage response, an observation largely at odds with the RNF4-findings with which this report conceptually fits. Overall I am supportive of publication of the manuscript however there is some confusion about how Atx3 fits into the DDR that needs clarification and one technical aspect that concerns me.”

We are pleased that the reviewer appreciates the novelty of our finding. Below we clarify how ataxin-3 fits into the DNA damage response and discuss also the technical aspects that were raised by the reviewer.

“Worryingly expression of WT-Atx3 doesn't fully rescue its depletion (~70% rescue. Figure 1C). This may be because over-expression is also deleterious to 53BP1 recruitment, or because the siRNAs hit another target. As the majority of the manuscript does not revisit complementation, but uses siRNA throughout there is a concern that some of the impacts seen are due to the non-specific knock-down. I would be reassured if the more vital findings were accompanied by complementation experiments (eg MDC1 turn-over, cell survival, NHEJ assays). (There is also no expression level to accompany Fig 1 C, although something similar is in fig 4C).”

We thank the reviewer for this suggestion. We also noticed that even though we reproducibly and significantly increase 53BP1 recruitment upon ectopic expression of GFP-tagged ataxin-3 in ATXN3-depleted cells, we never reached a full rescue. As the reviewer proposes, this may be due to the supraphysiological levels of the ectopically expressed ataxin-3. We have tested this hypothesis and indeed found that overexpression of ataxin-3 reduces 53BP1 recruitment (**Supplementary Fig. S7C; page 11:286-287**). The expression levels of the ectopically expressed wild-type and catalytically inactive ataxin-3 remaining after knock-down were still considerably higher than those observed for endogenous ataxin-3 (**Supplementary Fig. 7B; page 11:283-286**). Thus, overexpression of ataxin-3 provides a plausible explanation for the partial rescue observed by ectopically expressed wild-type ataxin-3.

To further validate the complementation and also the importance of the catalytic activity of ataxin-3, we have further analyzed the ability of wild-type and catalytically inactive ataxin-3 to rescue defects in the DSB response in ataxin-3-depleted cell. We show now that ubiquitylation of MDC1 (**Fig. 5C; page 9:234-238**) as well as the reduced recruitment of XRCC4 (**Fig. 8D; page 13:347-350**) and 53BP1 (**Fig. 6F; page 11:279-282**) can be partly rescued by expression of wild-type but not catalytic inactive GFP-ataxin-3.

“The model presented suggests an MDC1- SUMO-directed DUB regulates NHEJ, this needs more support.

If recruitment of Atx3 is PIAS4/Ubc9 dependent and also dependent on SIM3 of Atx3 then presumably interaction with MDC1 is also SUMO dependent. The immunoprecipitation (Fig 4A) therefore should show SUMO dependence. Could the authors repeat this experiment in the presence and absence of PIAS4 siRNA for example, or with the K1840R-MDC1 mutant - which is reported to be the main SUMO1-modification site on MDC1?”

We have tested the hypothesis of ataxin-3 being a MDC1-SUMO-directed DUB in detail by analyzing the interaction between MDC1 and ataxin-3 in Ubc9- or PIAS4-depleted cells (**Supplementary Fig. S4A; page 9:221-222**), as well as taking advantage of the MDC1 mutant that lacks the primary SUMOylation site K1840 (**Supplementary Fig. S4B; page 9:222-223**), as proposed by the reviewer. The combined results of these experiments strongly suggest that the interaction between MDC1 and ataxin-3 is constitutive and independent of SUMOylation. This suggests that the SUMO-dependent recruitment of ataxin-3 to DSBs is not facilitated by MDC1, but probably by another SUMOylated substrate(s) at DSBs. Consistent with this idea, we also found that the SUMO-mediated recruitment of ataxin-3 is not dependent on MDC1 (**Supplementary Fig. S4C; page 9:223-226**), as has been reported for RNF4 (Galanty et al, 2012). Thus, either the SUMOylation of another protein or protein group SUMOylation, which was shown to be required for an efficient DSB response (Psakhye & Jentsch, 2012), contributes to the recruitment of ataxin-3 to DSBs. It is, however, important to stress that our data unequivocally link SUMO and ataxin-3 since we show that 1) the recruitment of ataxin-3 is strictly dependent on SUMOylation (**Fig. 2H**), 2) ataxin-3 binds recombinant SUMO (**Fig. 3; Supplementary Fig. S2B**), 3) SUMO activates the catalytic activity of ataxin-3 (**Fig. 5E**), and 4) ataxin-3 regulates ubiquitylation of MDC1 (**Fig. 5B, C, D**). Thus, while we provide a large body of evidence in our study that connects ataxin-3 with SUMO and MDC1 at DSB, we have refrained in the revision from referring to ataxin-3 as a SUMO-targeted DUB.

“The relationship between ATx-3 and RNF4 (the presumed ligase it is countering) at the level of MDC1 would be expected to be key. Yet this has not been explored. Does reducing RNF4 levels reduce the levels of MDC1 ubiquitination in Atx3 depleted cells? Does loss of RNF4 restore normal early MDC1 turn-over-kinetics (as might be expected from the RNF168 data) what is the impact of RNF4 depletion on XRCC4? MDC1-K1840R would presumably be immune to both Atx3 and RNF4 - mediated regulations - confirming the role of this sites SUMO-modification as the likely route of regulation.”

We thank the reviewer for these suggestions. Indeed, our model suggested that ataxin-3 and RNF4 are acting at the level of ubiquitylated MDC1, which is modified in a SUMO-dependent manner requiring K1840 (Lou et al, 2006). Testing the effect of the MDC1-K1840R mutant would have required a cell line stably expressing siRNA-resistant version of wild-type or K1840R MDC1, which is not straightforward due to its large size (~2100 amino acids). In the revision we do, however, provide further support for this model by performing the experiments that were proposed by the reviewer. We now show that 1) depletion of RNF4 reduces the levels of ubiquitylated MDC1 in control and ataxin-3 depleted cells (**Fig. 5D; page 10:241-243**), 2) that depletion of RNF4 restores the residence time of MDC1 at the chromatin in ataxin-3-depleted cells (**Fig 4C, D; page 9:212-215**), 3) that depletion of RNF4 rescues the recruitment of XRCC4 to DSBs in ataxin-3-depleted cells (**Fig. 8E; page 13:351-357**). Finally, we show that co-depletion of ataxin-3 did not further compromise cell viability, but instead improved the ability of RNF4-depleted cells to survive DSBs induced by IR exposure, supporting antagonistic activities of ataxin-3 and RNF4 at the level of MDC1 (**Fig. 8F; page 14:364-370**).

“The link between increased MDC1 turn-over seen on Atx3 depletion at early time points and subsequent impact on DDR biology is difficult to see from the presented experiments. Why would the rapid re-association of MDC1 have a negative down-stream effect if other regulation is normal and MDC1 is not lost from lines of damage? Is the 'new' MDC1 not modified? What happens to RNF8 recruitment? How is the impact on RNF168 achieved - do the authors also think there is a non-MDC1 effect (say on H1-ubiquitination) ? Is this responsible for the NHEJ defect? How is the impact on XRCC4 achieved? Is this responsible for the NHEJ defect?”

For chromatin-associated events not only the steady-state amount of protein that localizes at the chromatin but also the timespan a single molecule resides at this location is of critical importance, as the latter determines the time window that the molecule has to set off a reaction. In the case of MDC1 direct downstream events that have to take place in order to trigger DNA damage-induced ubiquitylation are recruitment of RNF8 and ubiquitylation of RNF8 substrates, including histone H1 (Thorslund et al, 2015). If MDC1 dissociates before these events have taken place, the DNA damage response will be impaired. We speculate that the reduced residence time of MDC1 following ataxin-3 depletion is insufficient to efficiently trigger these downstream events. Consistent with our model, we found that all these events downstream of MDC1, such as DSB-induced ubiquitylation (**Fig. 6C**), and the recruitment of RNF8 (**Fig. 6A**), RNF168 (**Fig. 6B**), BRCA1 (**Fig. 6D**), and 53BP1 (**Fig. 6E**), are significantly affected by loss of ataxin-3. This is reminiscent of earlier findings showing that RNF4 regulates the chromatin retention time, but not the steady-state bound levels of MDC1 (Galanty et al, 2012), also hinting at the importance of regulating protein activity at the level of chromatin retention.

We have now tested the effect of MDC1 on DSB repair and found that MDC1 promotes 1) XRCC4 recruitment, and 2) XRCC4-mediated NHEJ of DSBs as measured using EJ5-GFP reporter and plasmid integration assays (**Fig. 8A-C; page 13:334-347**). Hence we believe that the reduced residence time of MDC1 is also responsible for the NHEJ defect observed in ataxin-3 depleted cells. Finally, we also show now that MDC1-dependent XRCC4 recruitment is subjected to opposite regulatory activities of RNF4 and ataxin-3. We discuss the connection between ataxin-3/RNF4 in regulating MDC1-dependent NHEJ at **page 17:440-455**.

“Functionally the model is not tied in. We might anticipate that RNF4 depletion in partially rescuing RNF168 (for whatever reason) may also restore some NHEJ and IR resistance of Atx3 depleted cells. Is this the case?”

In the revision, we now show that 1) RNF4 depletion partially rescues XRCC4 recruitment in ataxin-3 depleted cells (**Fig. 8D; page 13:347-350**), and 2) ataxin-3 depletion alleviates the sensitivity of RNF4-depleted cells to DSBs induced by IR (**Fig. 8F; page 14:364-370**), suggesting that RNF4 and ataxin-3 play antagonistic roles in DSB repair, likely by regulating the MDC1-RNF8/168 branch of the DNA damage response.

“An unexplored and facile possibility is that DDR protein expression is affected. A simple blot showing MDC1, RNF8, RNF4, RNF168, 53BP1 protein levels is needed to negate this possibility.”

We have tested the steady-state levels of the proposed proteins in U2OS cells that had been transfected with the two ataxin-3-specific siRNAs used in our studies (**Supplementary Fig. S3B; page 9:208-211; page 11:271-274**). We did not observe any changes in their steady-state levels with the exception of RNF168, which we found to be reduced upon knock-down with siATXN3-1, but not siATXN3-2. We would like to stress though that the effects of ataxin-3 depletion on RNF168 recruitment were also observed with siATXN3-2, which did not reduce the steady-state levels of RNF168. Moreover, the critical experiment in which we show that ataxin-3 and RNF4 have opposing activities on RNF168 recruitment was performed with siATXN3-2. Thus, our data exclude the possibility that the observed effects on MDC1, RNF8, RNF168, RNF4 and 53BP1 are caused by changes in their steady-state levels.

Referee #2

“In this study, Pfeiffer and co-workers identified a deubiquitylation enzyme (DUB), ataxin3, counteracts RNF4 and regulates the retention of MDC1. Ataxin3 functions in a SUMO dependent way. They found that depletion of ataxin3 could decrease the chromatin-dwell time of MDC1 at the damage sites and impairs the recruitment of the ubiquitin ligase RNF168. Depletion of ataxin3 also impairs NHEJ and sensitizes cells to IR. The authors proposed that the counteraction between ataxin3 and RNF4 provides a mechanism that could tightly regulates the MDC1 dependent signaling and DNA damage response. These findings are interesting and are likely to be of interest for a broad audience. However, the study needs to be strengthened with more evidence.”

We thank the reviewer for the helpful suggestions and are pleased to hear that the reviewer finds our findings interesting and appropriate for a broad audience.

“Major concerns:
 1. *The authors showed that ataxin3 binds to MDC1 in Figure 4A. Is MDC1 required for ATX3 recruitment? Does the SIM of ATX3 bind to sumoylated MDC1?”*

As discussed also above in response to the comments of reviewer #1, we have tested whether recruitment of ataxin-3 is dependent on MDC1. We show that ataxin-3 is still recruited to DSBs in the absence of MDC1 (**Fig. S4C; page 9:223-226**). Thus, the recruitment of ataxin-3 is dependent on SUMOylation, but independent of MDC1, which is similar to what has been observed for RNF4 (Galanty et al, 2012). We discuss this at **page 16:428-433**). As discussed above in response to reviewer #1, the interaction between MDC1 and ataxin-3 is independent of MDC1 SUMOylation, suggesting that the putative SIM is not required for the interaction.

*“2. In Figure 4, the authors should include cells rescued with WT/CA/*SIM mutants of ataxin3 in the experiment to confirm the deubiquitination of MDC1 is affected by ataxin3 and it depends on the SIMs.”*

As requested we do now show in the revision that the increased ubiquitylation of MDC1 in ataxin-3-depleted cells can be rescued by expressing wild-type ataxin-3, but not catalytically inactive ataxin-3 (**Fig. 5C; page 9:234-238**) but have been unable to test the importance of the SIM*. Despite various attempts we have been unable to generate a cell line stably expressing the ataxin-3 SIM* mutant, which would have been required to test the functional importance of the putative SIM in affecting MDC1's ubiquitylation status.

“3. Why RNF4 depletion reduced RNF168 accumulation?”

This is an interesting question. RNF4 is a pleiotropic ubiquitin ligase that targets a broad array of SUMOylated proteins. To this end, it is tempting to speculate that RNF4 may target SUMOylated proteins, such as MDC1 (Galanty et al, 2012; Lou et al, 2006; Yin et al, 2012) and RNF168 itself (Danielsen et al, 2012), to regulate RNF168 recruitment. However, neither we nor other groups have probed into this question and we feel that it would not be appropriate to speculate in this matter. Importantly, the effect that RNF4-depletion has on RNF168 accrual does not influence the main conclusions from our study and we feel that probing into the molecular mechanism underlying this phenomenon lies outside the scope of our study.

“4. As to the functional assay, the authors showed that depletion of ataxin3 affects the NHEJ. However, they did not show any data about HR. It would be better to check the effect on HR too. They should also check the recruitment of RNF8 and BRCA1.”

We have now analyzed the effect of ataxin-3-depletion on HR and show that this is indeed strongly affected in the DR-GFP reporter (**Fig. 7A; page 12:312-318**). Consistent with a defect in HR, we found that ataxin-3-depleted cells also display an increased sensitivity to PARP inhibitor (**Fig. 7D; page 13:329-332**). Mechanistically, we found that both the recruitment of RPA (**Fig. 7B; page 12:318-326**) and RAD51 (**Fig. 7C; page 13:327-329**) were impaired in ataxin-3-depleted cells, suggesting that the impaired HR response stems from a defect in the end-resection-dependent loading of the core HR factor RAD51.

We also found that knock-down of ataxin-3 results in a significant reduction in RNF8 recruitment (**Fig. 6A, page 10:268-272**), which is consistent with a defect early in the RNF8/RNF168 pathway. We also examined the accrual of BRCA1 to laser-inflicted damage (**Fig. 6D; page 11:276-279**) and ionizing radiation-induced foci (**Supplementary Fig. S6A; page 11:276-279**), and found this to be reduced in ataxin-3-depleted cells.

“Minor concerns:

1. *In abstract, the authors claimed that "Depletion of ataxin-3 increases the levels of ubiquitylated MDC1, which coincides with an increase in the chromatin-dwell time of MDC1 at DSBs". This is opposite what is reported in the text.”*

We thank the reviewer for pointing out this mistake. This has been corrected.

“2. Figure 1B, the authors should include a statistic analysis of the quantification.”

In the revision this panel is shown in **Supplementary Fig. 6A**. We have included statistical analysis showing that the effects are highly significant.

“3. Figure 1C, the authors should show a representative picture of the experiment.”

In the revision, this panel is shown in **Figure 6F**. Representative images of the experiment are now shown in **Supplementary Fig. S7A**.

“4. In the text, from line 127 to line 141, all the Fig2 E,F,G are mis-referred.”

We thank the reviewer for pointing out this mistake. This has been corrected. The respective panels are now shown **Fig. 1E, F and G**.

Referee #3

“Pfeiffer et al. reported a deubiquitylation enzyme (DUB) ataxin-3 involved in DNA double strand break (DSB) response.

In this manuscript, authors showed that ataxin-3 was recruited to DSB induced by laser micro-irradiation or bleomycin treatment. The retention of Ataxin-3 to DSBs was reported to depend on SUMO binding rather than its ubiquitin interaction motif (UIM) and DUB activity. They suggested Ataxin-3 enhanced the assembly of 53bp1 to DSB by regulating chromatin retention of MDC1, therefore supported NHEJ repair.

Much of the data presented to support the contentions above is marginal and not very convincing. A case in point being Fig 4 where the differences in the retention time on chromatin are small. While ataxin-3 may indeed be recruited to DSBs this is not really new, but is consistent with a previously published report (Nishi et al., 2014).”

Using FRAP-FLIP in combination with laser micro-irradiation, we found that after 50 seconds there was an approximate 3-fold decrease in the residence time of MDC1 at DSBs in ataxin-3 depleted cells (**Fig. 4B, C, D; $T_{1/2}$ control = 30 seconds, $T_{1/2}$ siATX3 = 50 seconds**). We have now included a histogram plot to highlight this difference in chromatin dwell-time upon knock-down of ataxin-3 (**Fig. 4D**). The decrease in MDC1 steady-state bound levels at DSBs is indeed quite small, but this is not unexpected since the presence of unrepaired DSBs in the laser micro-irradiated region will result in the continuous recruitment of new MDC1 molecules from the nucleoplasm. Efficient recruitment of DNA repair enzymes with high-steady state bound levels at sites of DNA damage has been observed even when DNA repair was abortive and impaired (Giglia-Mari et al, 2006; Luijsterburg et al, 2010). Thus, recruitment of a DSB repair protein by itself is no proof of functional DNA damage signaling and repair. It is noteworthy that in earlier studies a similar quantitative effect on MDC1 residence time was observed in RNF4-depleted cells with no net effect on MDC1’s steady-state bound levels at DSBs, yet with strong functional consequences for the DSB response, leaving little doubt about the physiological significance of this phenomenon (Galanty et al, 2012; Luo et al, 2012; Yin et al, 2012). In the revision we further strengthen the functional link between ataxin-3 and RNF4 by showing that depletion of RNF4 restores the residence time of MDC1 in ataxin-3-depleted cells (**Fig. 4C, D; page 9:211-215**), and by demonstrating that ataxin-3 and RNF4 play antagonistic roles in regulating the MDC1-dependent signaling (**Fig. 6H**) and repair of DSBs (**Fig. 8E, F**).

“What is new here is the idea that ataxin-3 is recruited to DSBs by interaction with SUMO. This is based on knocking down SUMO conjugating enzymes and mutation of the putative SUMO interaction motif in ataxin-3.”

Our finding that the recruitment of ataxin-3 is dependent on SUMOylation is indeed novel. Importantly, in the revision, we have further established the link between ataxin-3 and SUMO by showing *1)* that SUMO1 interacts with recombinant ataxin-3 (**Fig. 3A and Supplementary Fig. S2B; page 7:154-161**), *2)* that the catalytic domain and not its C-terminal part is important for the interaction (**Fig. 3B,C; page 7:161-164**), *3)* that SUMO stimulates the catalytic activity of ataxin-3 in vitro (**Fig 5E; page 10:247-255**), and *4)* that ataxin-3 counteracts the SUMO-targeted ubiquitin ligases RNF4 (**Fig. 4D; Fig. 6H; Fig. 8E, F**). In our opinion the real novelty of our findings lies in

the fact that we have identified a DUB that counteracts RNF4 in regulating the signaling and repair of DSBs by SUMO-targeted ubiquitylation. In order to better emphasize the main message of our study, we have changed the title of the manuscript to “*Ataxin-3 consolidates the MDC1-dependent DNA double-strand break response by counteracting the SUMO-targeted ubiquitin ligase RNF4*”.

“However a large number of proteins at DSBs are SUMO modified and the authors did not present evidence that the protein containing the SIM mutant was functional. The pull downs from cell extracts are not really informative as we have no idea of how direct the interact is.”

To address this issue, we have performed experiments with recombinant ataxin-3 and recombinant SUMO1 conjugated to beads. These experiments show that ataxin-3 directly interacts with SUMO1 (**Supplementary Fig S2B; page 7:158-161**). Unfortunately, while we could purify ataxin-3 wild-type, we were unable to purify the SIM* mutant, which could therefore not be included in this analysis. However, we do show that not the C-terminal, but the SIM-containing N-terminal region of ataxin-3 interacts with SUMO1. This is in agreement with results from our pull down experiments showing that wild-type ataxin-3, but not the ataxin-3 SIM* mutant interacts with SUMO1. Together, our findings suggest functional relevance for ataxin-3's SIM in SUMO binding, which is in line with the fact that the SIM in ataxin-3 has been identified *in silico* in an earlier study (Guzzo & Matunis, 2013).

“What is missing from this paper is a direct demonstration that ataxin-3 is indeed a SUMO targeted DUB. This requires an *in vitro* experiment with purified recombinant ataxin-3 and a variety of recombinant substrates that are both SUMO modified and ubiquitinated. Such substrates could include ubiquitinated SUMO chains that are the product of RNF4 action or a substrate that is modified at different sites by SUMO and ubiquitin. The defining feature of a SUMO targeted DUB is that DUB activity should be enhanced by SUMO modification and reduced by removal of SUMO. In fact this would be manifest in a reduced *K_m* for substrate in the presence of SUMO. Without such data the statement that ataxin-3 is a SUMO targeted deubiquitination enzyme cannot be justified.”

Even though we conclusively show that ataxin-3 is targeted to DSBs by SUMO, we agree that in order to define ataxin-3 as a SUMO-targeted DUB it will be important to show that it is also targeted to specific substrates in a SUMO-dependent fashion. The proposed experiments are however technically challenging. For example, ataxin-3 is known to act preferentially on longer K63-linked ubiquitin chains and it has proven to be difficult to generate ubiquitylated SUMO with ubiquitin chains of the desired length (**Dr. Alfred Vertegaal, personal communication**). However, inspired by the fact that our data suggest that SUMO interacts with a putative SIM localized within the catalytic domain of ataxin-3, we have instead explored a possible effect of SUMO on the deubiquitylating activity of ataxin-3. Interestingly, we found that pre-incubation of recombinant ataxin-3 with recombinant SUMO1 resulted in a significant increase in its activity towards ubiquitin chains (**Fig 5E; page 10:247-255**). These results suggest that ataxin-3 is a DUB whose activity is enhanced by the presence of SUMO, strengthening the link and revealing an unprecedented crosstalk between SUMO and ataxin-3. We understand that referring to ataxin-3 as a SUMO-targeted DUB may be misunderstood. To avoid confusion and based on our new results, we have therefore removed any reference to ataxin-3 as a SUMO-targeted DUB in our revised manuscript.

References

- Danielsen JR, Povlsen LK, Villumsen BH, Streicher W, Nilsson J, Wikstrom M, Bekker-Jensen S, Mailand N (2012) DNA damage-inducible SUMOylation of HERC2 promotes RNF8 binding via a novel SUMO-binding Zinc finger. *J Cell Biol* **197**: 179-187
- Galanty Y, Belotserkovskaya R, Coates J, Jackson SP (2012) RNF4, a SUMO-targeted ubiquitin E3 ligase, promotes DNA double-strand break repair. *Genes Dev* **26**: 1179-1195
- Giglia-Mari G, Miquel C, Theil AF, Mari PO, Hoogstraten D, Ng JM, Dinant C, Hoeijmakers JH, Vermeulen W (2006) Dynamic interaction of TTDA with TFIID is stabilized by nucleotide excision repair in living cells. *PLoS Biol* **4**: e156
- Guzzo CM, Matunis MJ (2013) Expanding SUMO and ubiquitin-mediated signaling through hybrid SUMO-ubiquitin chains and their receptors. *Cell Cycle* **12**: 1015-1017

Lou Z, Minter-Dykhouse K, Franco S, Gostissa M, Rivera MA, Celeste A, Manis JP, van Deursen J, Nussenzweig A, Paull TT, Alt FW, Chen J (2006) MDC1 maintains genomic stability by participating in the amplification of ATM-dependent DNA damage signals. *Mol Cell* **21**: 187-200

Luijsterburg MS, von Bornstaedt G, Gourdin AM, Politi AZ, Mone MJ, Warmerdam DO, Goedhart J, Vermeulen W, van Driel R, Hofer T (2010) Stochastic and reversible assembly of a multiprotein DNA repair complex ensures accurate target site recognition and efficient repair. *J Cell Biol* **189**: 445-463

Luo K, Zhang H, Wang L, Yuan J, Lou Z (2012) Sumoylation of MDC1 is important for proper DNA damage response. *EMBO J* **31**: 3008-3019

Psakhye I, Jentsch S (2012) Protein group modification and synergy in the SUMO pathway as exemplified in DNA repair. *Cell* **151**: 807-820

Thorslund T, Ripplinger A, Hoffmann S, Wild T, Uckelmann M, Villumsen B, Narita T, Sixma TK, Choudhary C, Bekker-Jensen S, Mailand N (2015) Histone H1 couples initiation and amplification of ubiquitin signalling after DNA damage. *Nature* **527**: 389-393

Yin Y, Seifert A, Chua JS, Maure JF, Golebiowski F, Hay RT (2012) SUMO-targeted ubiquitin E3 ligase RNF4 is required for the response of human cells to DNA damage. *Genes Dev* **26**: 1196-1208

2nd Editorial Decision

11 January 2017

Thank you for submitting your revised manuscript on Ataxin-3 in the DNA DSB response for our editorial consideration. Two of the original referees have now once more assessed the study in depth, and I am pleased to inform you that they both consider the manuscript significantly improved and most of the key concerns adequately addressed. Pending further modification of a few minor issues (one missing control and some additional discussion), we should therefore be happy to accept the manuscript for publication in The EMBO Journal. I am therefore returning the study to you for a final round of revision in order to allow you to incorporate these last changes.

REFeree REPORTS

Referee #1:

The authors have made considerable efforts to answer the questions of reviewers. As a result the manuscript offers a clearer view of the role of Atx3 in countering the function of RNF4 in the DNA DSB response and in particular at the level of the MDC1 substrate. The authors have enough information to pull back from naming Atx3 a SUMO-targeting DUB, but the elements of SUMO regulation on Atx3 nevertheless maintain considerable novelty.

There are areas of incongruity and incompleteness - which are largely pointed out by the authors and await future work;

For example In cell survival terms (in response to IR) on Atx3 depletion (which results in a quicker MDC1 processing) is not ameliorated by RNF4 co-depletion - which is shown to restore slower MDC1 processing, which is surprising and counter to expectations from the data in the manuscript. This is particularly in view that reduced survival on RNF4 loss (which alone radically reduces MDC1 clearance from damaged DNA) is ameliorated by Atx 3 loss (as expected since this would restore Ub-MDC1). Why the same relationship is not the case for both depletions is at odds with the observed impacts on MDC1. This perhaps suggests another role for Atx3 in the DDR?

It's still not entirely clear what the down-stream defect is on Atx3 loss. Yes it is clear that RNF8 recruitment is somewhat reduced (not by a great deal) and RNF168 is more affected but why? The steady-state recruitment of MDC1 is not affected by Atx3 loss even if residency of single molecules is, evidence to show the down-stream impacts of Atx3 loss are MDC1-mediated is missing - I take

the point that a model can nevertheless be speculated from the observations.

Taken together this story brings new insight into regulation of RNF4-mediated steps in the signaling and repair of DNA DSBs.

Referee #2:

The authors addressed most of my concerns. I support the publication of this manuscript pending one minor point. In response to my Point 1, the authors did MDC1 knockdown to show that the recruitment of Axaxin-3 is independent of MDC1. However, no Western blot show the knockdown effect. It is important to show this to make sure the negative result was not due to insufficient knockdown.

2nd Revision - authors' response

02 February 2017

Response to Referee #1:

The authors have made considerable efforts to answer the questions of reviewers. As a result the manuscript offers a clearer view of the role of Atx3 in countering the function of RNF4 in the DNA DSB response and in particular at the level of the MDC1 substrate. The authors have enough information to pull back from naming Atx3 a SUMO-targeting DUB, but the elements of SUMO regulation on Atx3 nevertheless maintain considerable novelty. There are areas of incongruity and incompleteness - which are largely pointed out by the authors and await future work;

For example In cell survival terms (in response to IR) on Atx3 depletion (which results in a quicker MDC1 processing) is not ameliorated by RNF4 co-depletion - which is shown to restore slower MDC1 processing, which is surprising and counter to expectations from the data in the manuscript. This is particularly in view that reduced survival on RNF4 loss (which alone radically reduces MDC1 clearance from damaged DNA) is ameliorated by Atx 3 loss (as expected since this would restore Ub-MDC1). Why the same relationship is not the case for both depletions is at odds with the observed impacts on MDC1. This perhaps suggests another role for Atx3 in the DDR?

This point is well taken and we agree that ataxin-3 may have additional roles in the DNA damage response. We comment on this specific result now on page 15.

It's still not entirely clear what the down-stream defect is on Atx3 loss. Yes it is clear that RNF8 recruitment is somewhat reduced (not by a great deal) and RNF168 is more affected but why? The steady-state recruitment of MDC1 is not affected by Atx3 loss even if residency of single molecules is, evidence to show the down-stream impacts of Atx3 loss are MDC1- mediated is missing - I take the point that a model can nevertheless be speculated from the observations.

In the discussion we speculate that the chromatin retention time of RNF8 may be reduced similar as what we observed for MDC1 and that this may result in inefficient ubiquitylation of histone H1 and impaired recruitment of RNF168.

Taken together this story brings new insight into regulation of RNF4-mediated steps in the signaling and repair of DNA DSBs.

Response to Referee #2:

The authors addressed most of my concerns. I support the publication of this manuscript pending one minor point. In response to my Point 1, the authors did MDC1 knockdown to show that the recruitment of Axaxin-3 is independent of MDC1. However, no Western blot shows the knockdown effect. It is important to show this to make sure the negative result was not due to insufficient knockdown.

We show now in Appendix Fig S4C a Western blot showing that MDC1 was efficiently depleted in this experiment.

3rd Editorial Decision

06 February 2017

Thank you for submitting your final revised manuscript for our consideration. I am pleased to inform you that we have now accepted it for publication in The EMBO Journal.

Corresponding Author Name: Haico van Attikum, Nico P. Dantuma

Journal Submitted to: The EMBO journal

Manuscript Number: EMBOJ-2016-95151R